# Adipocytic sclerostin loop3-LRP4 interaction required by sclerostin to impair whole-body lipid and glucose metabolism

Hewen Jiang[1,10], Xiaohui Tao[2,3,4,10], Sifan Yu [2,3,4,10], Yihao Zhang[1,10], Yuan Ma[1,10], Nanxi Li[2,3,4], Shenghang Wang[2,3,4], Ning Zhang[1], Xin Yang[2,3,4], Shijian Ding[2,3,4], Chuanxin Zhong [2,3,4], Haitian Li[2,3,4], Zhanghao Li[2,3,4], Xiaoxin Wen[2,3,4], Huarui Zhang[1], Zefeng Chen[2,3,4], Meiheng Sun[2,3,4], Hang Luo[1], Meishen Ren[2,3,4], Chongguang Lei[1], Yuanyuan Yu [2,3,4], Jin Liu [2,3,4], Zongkang Zhang [1], Aiping Lyu [2,3,4], Hui Sheng [5] ✉, Dijie Li [2,3,4,6,7,8,9] ✉, Luyao Wang [2,3,4] ✉, Ge Zhang [2,3,4] ✉ & Bao-Ting Zhang [1] ✉

Sclerostin, which has three loops, inhibits bone formation and impairs whole-body lipid and glucose metabolism. The marketed therapeutic sclerostin antibody for postmenopausal osteoporosis (POP) mainly targeting loop2 promotes bone formation and improves whole-body lipid and glucose metabolism. However, FDA/EMA warns of its cardiovascular risk. We previously demonstrate that sclerostin loop3 contributes to the inhibitory effect of sclerostin on bone formation but not its cardioprotective effect. Here we find elevated serum sclerostin levels in both POP-T2DM patients and newly-diagnosed T2DM patients and further demonstrate that sclerostin loop3 participates in the impairment effect of sclerostin on whole-body lipid and glucose metabolism in vivo. Mechanistically, specific blockade of adipocytic sclerostin loop3-LRP4 interaction attenuates the impairment effect of sclerostin on lipid and glucose metabolism in vitro and in vivo. This study provides an innovative strategy, blocking adipocytic sclerostin loop3-LRP4 interaction, to normalize lipid and glucose metabolism in POP-T2DM patients, in cardiovascular safety.

Osteoporosis commonly coexists with type 2 diabetes mellitus (T2DM) in postmenopausal women[1–3]. In a study of postmenopausal osteoporosis (POP) patients, it was found that the serum sclerostin levels were significantly higher in POP patients coexisting with T2DM compared to POP patients without T2DM[4]. Sclerostin plays important roles in inhibiting bone formation[5–7] and protecting cardiovascular system[8,9]. Intriguingly, sclerostin was also found to impair whole-body lipid and glucose metabolism[10,11]. Sclerostin has three loops, including loop1, loop2, and loop3[12]. The marketed therapeutic sclerostin antibody mainly targeting sclerostin loop2, could not only promote bone

[1]School of Chinese Medicine, The Chinese University of Hong Kong, Hong Kong, China. [2]Guangdong-Hong Kong Macao Greater Bay Area International Research Platform for Aptamer-Based Translational Medicine and Drug Discovery, Hong Kong, China. [3]Law Sau Fai Institute for Advancing Translational Medicine in Bone & Joint Diseases, School of Chinese Medicine, Hong Kong Baptist University, Hong Kong, China. [4]Institute of Integrated Bioinformedicine and Translational Science, School of Chinese Medicine, Hong Kong Baptist University, Hong Kong, China. [5]Osteoporosis and Sarcopenia Center, Department of Endocrinology and Metabolism, School of Medicine, Tongji University, The Shanghai Tenth People's Hospital, Shanghai, China. [6]Guangxi Universities Key Laboratory of Stem cell and Biopharmaceutical Technology, College of Life Sciences, Guangxi Normal University, Guilin, Guangxi, China. [7]Shenzhen Institute for Research and Continuing Education (IRACE), Hong Kong Baptist University, Shenzhen, Guangdong, China. [8]Research Center for Biomedical Sciences, Guangxi Normal University, Guilin, Guangxi, China. [9]Key Laboratory of Ecology of Rare and Endangered Species and Environmental Protection (Ministry of Education), Guangxi Normal University, Guilin, Guangxi, China. [10]These authors contributed equally: Hewen Jiang, Xiaohui Tao, Sifan Yu, Yihao Zhang, Yuan Ma. ✉e-mail: shenghui@tongji.edu.cn; lidijie@gxnu.edu.cn; luyaowang@hkbu.edu.hk; zhangge@hkbu.edu.hk; zhangbaoting@cuhk.edu.hk

formation to reverse established osteoporosis[13–15], but also normalize whole-body lipid and glucose metabolism in high-fat diet (HFD)-induced mice[10]. However, due to the reported severe cardiac ischemic events, both U.S. Food and Drug Administration (US-FDA) and European Medicines Agency (EMA) raised the warning of its cardiovascular risk in POP patients[16]. Therefore, it is desirable to develop a new generation of sclerostin inhibitors without increasing cardiovascular risk for not only promoting bone formation, but also normalizing lipid and glucose metabolism in POP patients coexisting with T2DM.

Our retrospective study found that serum sclerostin levels were significantly higher in POP patients coexisting with T2DM compared to POP patients without T2DM. Consistently, the ovariectomized (OVX) mice fed a HFD also showed higher serum sclerostin levels than those fed a chow diet. Surprisingly, the serum sclerostin levels were significantly elevated in newly diagnosed T2DM patients in our large-sample follow-up analysis. Sclerostin has emerged as a novel target for promoting bone formation and normalizing lipid and glucose metabolism, but with cardiovascular safety concern[10,17,18]. In our previous studies, we found that sclerostin loop3 contributed to the inhibitory effect of sclerostin on bone formation, whereas the protective effect of sclerostin on cardiovascular system was independent of loop3[19–21]. A sclerostin loop3-specific aptamer (Apc001) developed by our group demonstrated no cardiovascular concerns in mice and could promote bone formation to reverse established osteoporosis in OVX rats and osteogenesis imperfecta mice[19,20,22]. Thus, it is of great interest to investigate whether and how sclerostin loop3 participates in the impairment effect of sclerostin on whole-body lipid and glucose metabolism, which is a hurdle to develop a precise therapeutic sclerostin inhibition strategy for not only promoting bone formation, but also normalizing lipid and glucose metabolism without cardiovascular risk.

In this study, we investigated the role of sclerostin loop3 in the impairment effect of sclerostin on whole-body lipid and glucose metabolism using both genetic and pharmacologic approaches. Then, we investigated whether genetic loop3 mutation or pharmacologic loop3 inhibition could attenuate the HFD-induced whole-body lipid and glucose metabolism disorders. To mechanistically reveal how sclerostin loop3 participated in the impairment effect of sclerostin on whole-body lipid and glucose metabolism, we performed protein-protein interaction assays to identify a receptor of sclerostin loop3 in adipocytes. Further, we investigated whether the adipocytic sclerostin loop3-receptor (the low-density lipoprotein receptor-related protein 4, LRP4) interaction was required by sclerostin to impair whole-body lipid and glucose metabolism in vitro and in vivo.

## Results

### Serum sclerostin levels were significantly higher in newly diagnosed T2DM (ND-T2DM) patients, POP patients with T2DM, and corresponding animal models

From our follow-up community-based retrospective study over 5 years, it was found that the serum sclerostin levels were significantly higher in the individuals newly diagnosed T2DM (ND-T2DM) when compared to their respective baselines (Fig. 1a). The longitudinal data suggested that serum sclerostin levels could be a biomarker for the development of T2DM. From the other retrospective study, we found that serum sclerostin levels were significantly higher in POP patients with T2DM ($n = 24$) compared to POP patients without T2DM ($n = 22$) (Fig. 1b). Meanwhile, the serum free fatty acids (FFA), triglycerides, glycosylated hemoglobin (HbA1c) and fasting blood glucose (FBG) levels were significantly higher in POP patients with T2DM compared to POP patients without T2DM (Supplementary Table 1). The cross-sectional data suggested the correlation between serum sclerostin levels and T2DM (lipid and glucose metabolism disorders) in POP patients.

To examine the correlation between serum sclerostin levels and lipid and glucose metabolism disorders in the postmenopausal osteoporosis mouse model, 10-week-old mice (2 weeks after ovariectomy or Sham surgery) were induced by HFD for 12 weeks. In accordance with previously published data of mice with HFD induction[10], the serum sclerostin levels were significantly higher in Sham + HFD group than those in Sham + Chow group, and in particular, the serum sclerostin levels were significantly higher in OVX + HFD group than those in OVX + Chow group (Supplementary Fig 1a). The OVX + HFD group and Sham + HFD group showed higher body weights, fat pad weights, and FFA levels when compared to OVX + Chow group and Sham + Chow group, respectively (Supplementary Fig 1b-d). In line with these data, the FBG levels in HFD-fed groups were elevated as compared to their corresponding chow-fed groups (Supplementary Fig 1e). The data from the glucose tolerance test (GTT) and insulin tolerance test (ITT) further demonstrated a significant impairment in glucose tolerance and insulin sensitivity in the groups fed a HFD, in comparison to corresponding groups received a chow diet (Supplementary Fig 1f). All the above findings further indicated the correlation between the elevated serum sclerostin levels and impaired

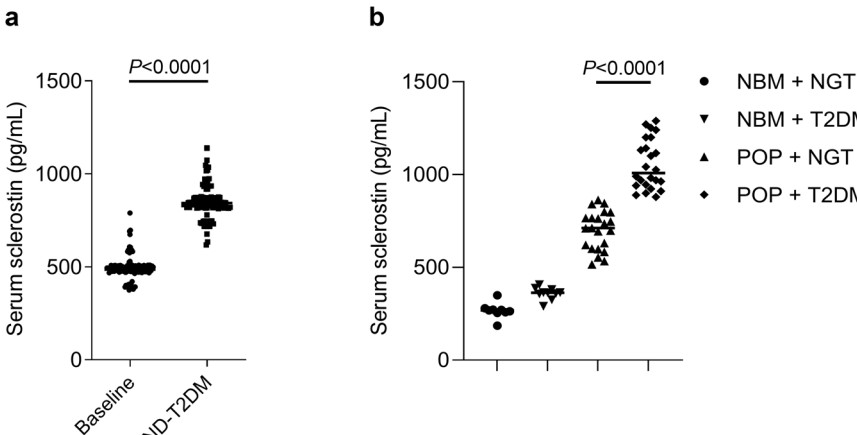

**Fig. 1 | The serum sclerostin levels were associated with type 2 diabetes mellitus (T2DM). a** The serum sclerostin levels in individuals before (Baseline) and after newly diagnosed T2DM (ND-T2DM). Statistical significance was calculated using paired *t*-test. **b** The serum sclerostin levels in postmenopausal osteoporosis (POP) patients with T2DM and POP patients without T2DM. Statistical significance was calculated using unpaired *t*-test. Note: Baseline: healthy individuals ($n = 119$); ND-T2DM: newly diagnosed T2DM patients ($n = 119$). Normal bone mass (NBM), normal glucose tolerance (NGT). Individuals without T2DM ($n = 9$); individuals with T2DM ($n = 10$); POP patients without T2DM ($n = 22$); POP patients with T2DM ($n = 24$). All data were expressed as mean ± SD. All tests were two-sided.

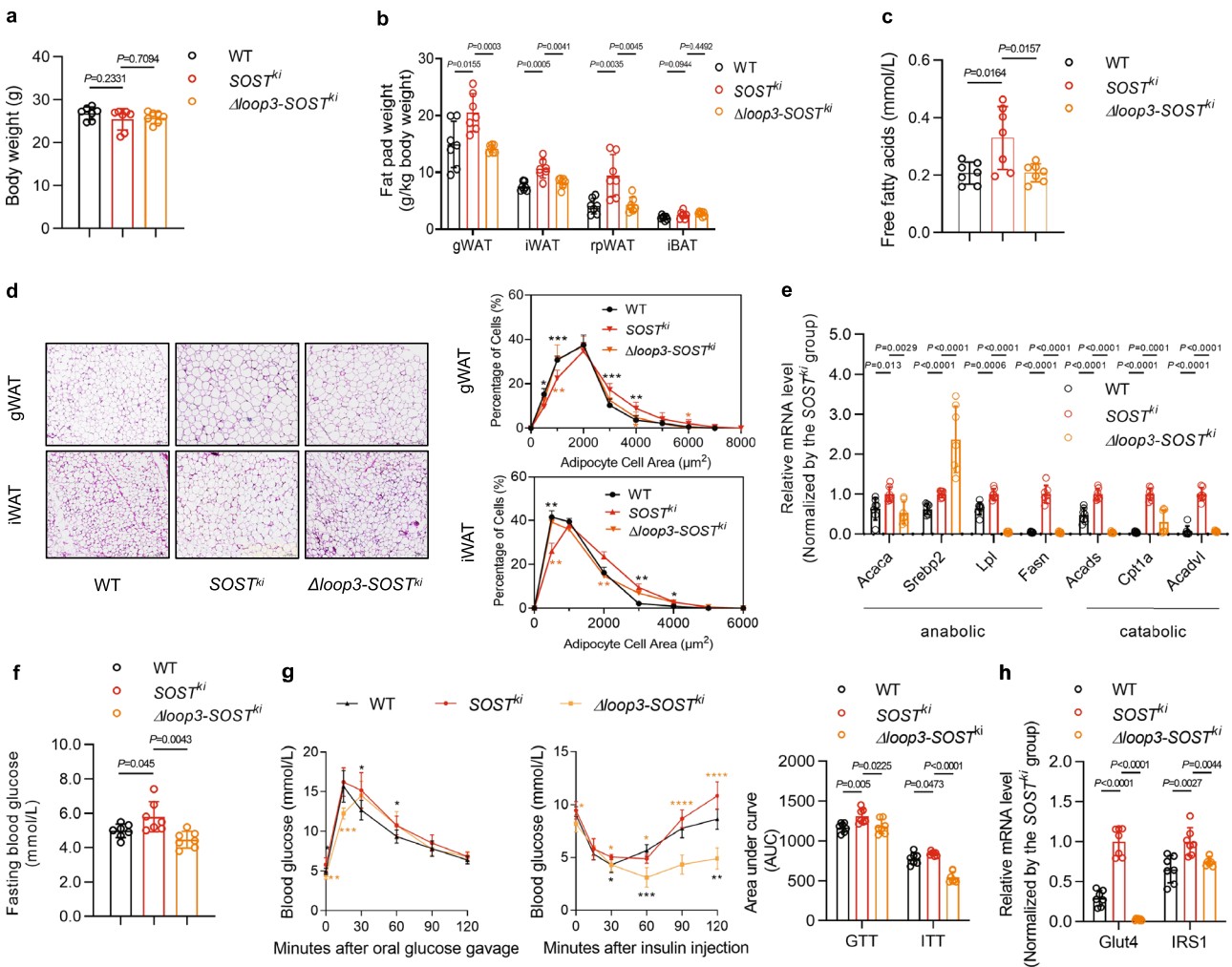

**Fig. 2 | Loop3 deficiency by genetic truncation attenuated the impairment effect of sclerostin on whole-body lipid and glucose metabolism in vivo. a** Body weights of wild-type (WT) mice, full-length sclerostin knock-in (*SOST^{ki}*) mice, and loop3-deficient sclerostin knock-in (*Δloop3-SOST^{ki}*) mice. **b** Fat pad weights in WT mice, *SOST^{ki}* mice, and *Δloop3-SOST^{ki}* mice. gWAT: gonadal white adipose tissue; iWAT: inguinal white adipose tissue; rpWAT: retroperitoneal white adipose tissue; iBAT: interscapular brown adipose tissue. **c** Serum free fatty acids in WT mice, *SOST^{ki}* mice, and *Δloop3-SOST^{ki}* mice. **d** Representative images of histological sections (left) and frequency distribution of adipocyte sizes (right) in gWAT and iWAT from WT mice, *SOST^{ki}* mice, and *Δloop3-SOST^{ki}* mice. Scale bars, 100 μm. (**e**) Expression levels of genes associated with lipid anabolism (*Acaca, Srebp2, Lpl*, and

*Fasn*) and catabolism (*Acads, Cpt1a* and *Acadvl*) in gWAT from WT mice, *SOST^{ki}* mice and *Δloop3-SOST^{ki}* mice detected by qPCR. **f** Fasting blood glucose in WT mice, *SOST^{ki}* mice, and *Δloop3-SOST^{ki}* mice. **g** Glucose tolerance test (GTT) (left), insulin tolerance test (ITT) (middle), and area under the curve (AUC) analysis for GTT and ITT (right) in WT mice, *SOST^{ki}* mice, and *Δloop3-SOST^{ki}* mice. **h** Expression levels of genes associated with glucose metabolism in gWAT from WT mice, *SOST^{ki}* mice, and *Δloop3-SOST^{ki}* mice detected by qPCR. Note: n = 7 biologically independent samples per group. All data were expressed as mean ± SD. *$P < 0.05$, **$P < 0.01$, ***$P < 0.001$ and ****$P < 0.0001$ for intergroup comparison (black *: WT vs. *SOST^{ki}*; orange *: *SOST^{ki}* vs. *Δloop3-SOST^{ki}*). Statistical significance was calculated using unpaired *t*-test. ns: no significance. All tests were two-sided.

whole-body lipid and glucose metabolism in the postmenopausal osteoporosis mouse model.

## Loop3 deficiency by genetic truncation attenuated the impairment effect of sclerostin on whole-body lipid and glucose metabolism in vivo

Since sclerostin loop3 has been found to play a critical role in the inhibitory effect of sclerostin on bone formation but not the cardiovascular protective action of sclerostin, the correlation between serum sclerostin and whole-body lipid and glucose metabolism disorders in both human and animal models prompted us to further examine whether sclerostin loop3 could contribute to the impairment effect of sclerostin on whole-body lipid and glucose metabolism. The full-length sclerostin knock-in (*SOST^{ki}*) mice and loop3-deficient sclerostin knock-in (*Δloop3-SOST^{ki}*) mice were constructed (Supplementary Fig 2a, b). Higher serum sclerostin levels were found in both 3-month-old *SOST^{ki}* and *Δloop3-SOST^{ki}* mice than those in wild-type (WT) mice

(Supplementary Fig 3a). Consistent with a previous report[10], our data from the *SOST^{ki}* mice and WT mice showed that sclerostin imposed an impairment effect on whole-body lipid and glucose metabolism in vivo (Fig. 2). Both the *SOST^{ki}* mice and *Δloop3-SOST^{ki}* mice had normal body weight (Fig. 2a). The fat pad weights and serum FFA levels were significantly higher in *SOST^{ki}* mice than WT mice, and intriguingly, the fat pad weights and serum FFA levels were significantly lower in *Δloop3-SOST^{ki}* mice than *SOST^{ki}* mice (Fig. 2b, c). In both the gonadal white adipose tissue (gWAT) and inguinal white adipose tissue (iWAT), the adipocyte sizes were markedly larger in *SOST^{ki}* mice than WT mice, which were markedly smaller in *Δloop3-SOST^{ki}* mice than *SOST^{ki}* mice (Fig. 2d). The expression levels of genes associated with lipid anabolism (*Acaca, Srebp2, Lpl* and *Fasn*) and lipid catabolism (*Acads, Cpt1a* and *Acadvl*) in gWAT and iWAT of *SOST^{ki}* mice were significantly higher than WT mice, while *Δloop3-SOST^{ki}* mice showed significantly lower expression levels of the above genes than *SOST^{ki}* mice, including *Acaca, Lpl, Fasn, Acads, Cpt1a* and *Acadvl* (Fig. 2e, Supplementary

Fig 3b). It implied that sclerostin knockin might increase the expression levels of genes related to both lipid anabolism and lipid catabolism simultaneously for the regulation of lipid and glucose metabolism in vivo. Moreover, it was found that FBG was higher in $SOST^{ki}$ mice than WT mice, which was lower in $\Delta loop3\text{-}SOST^{ki}$ mice than $SOST^{ki}$ mice (Fig. 2f). The glucose tolerance and insulin sensitivity in $SOST^{ki}$ mice were impaired relative to WT mice, while $\Delta loop3\text{-}SOST^{ki}$ mice exhibited improvements in glucose tolerance and insulin sensitivity during GTT and ITT relative to $SOST^{ki}$ mice (Fig. 2g). Consistently, we observed a higher phosphorylation level of IRS1 and a lower phosphorylation level of AKT in gWAT of $SOST^{ki}$ mice than WT mice, while $\Delta loop3\text{-}SOST^{ki}$ mice showed a lower phosphorylation level of IRS1 and a higher phosphorylation level of AKT than $SOST^{ki}$ mice (Supplementary Fig 3c). Thereafter, we examined the effect of loop3 deficiency on the mRNA expression levels of the gene encoding glucose transporter member 4 (Glut4) and the gene involved in Glut4-mediated insulin-dependent glucose uptake (IRS1, insulin receptor substrate 1). Consistently, the expression levels of genes associated with glucose metabolism (Glut4 and IRS1) in both gWAT and iWAT of $SOST^{ki}$ mice were significantly higher when compared to those of WT mice, but $\Delta loop3\text{-}SOST^{ki}$ mice showed significantly lower expression levels of these genes when compared to those of $SOST^{ki}$ mice, which was suggestive of the importance of loop3 in sclerostin-promoted glucose uptake into adipose tissues in vivo (Fig. 2h and Supplementary Fig 3d). The above genetic evidence indicated that loop3 in sclerostin played an important role in the impairment effect of sclerostin on whole-body lipid and glucose metabolism in vivo.

## Loop3 inhibition by the pharmacologic inhibitor Apc001OA (a sclerostin loop3-specific aptamer) attenuated the impairment effect of sclerostin on whole-body lipid and glucose metabolism in vivo

Since sclerostin exhibited an impairment effect on whole-body lipid and glucose metabolism as mentioned above, we next pharmacologically investigated whether inhibition of sclerostin loop3 could attenuate the impairment effect of sclerostin on whole-body lipid and glucose metabolism in vivo. A chemically modified sclerostin loop3-specific aptamer (Apc001OA, Apc001 with octadecanedioic acid modification) with extended half-life developed by our group was used as a pharmacologic inhibitor[19,20,23,24]. In 3-month-old $SOST^{ki}$ mice treated with Apc001OA ($SOST^{ki}$ + Apc001OA) for 12 weeks (Fig. 3a), the serum sclerostin levels were comparable to those in $SOST^{ki}$ mice treated with Vehicle ($SOST^{ki}$ + Veh) (Supplementary Fig 4a). There was no significant difference in body weights between $SOST^{ki}$ + Apc001OA group and $SOST^{ki}$ + Veh group (Fig. 3b). It was noted that the fat pad weights were significantly lower in $SOST^{ki}$ + Apc001OA group compared to $SOST^{ki}$ + Veh group (Fig. 3c), while there was no difference in the serum FFA levels (Fig. 3d). The adipocyte sizes in both gWAT and iWAT of $SOST^{ki}$ + Apc001OA group were smaller than those in $SOST^{ki}$ + Veh group (Fig. 3e). Intriguingly, the expression levels of genes associated with lipid anabolism (Acaca, Srebp2, Lpl and Fasn) and lipid catabolism (Acads, Cpt1a and Acadvl) in gWAT and iWAT of $SOST^{ki}$ + Apc001OA group were significantly lower, when compared to those of $SOST^{ki}$ + Veh group (Fig. 3f and Supplementary Fig 4b). This might be attributed to higher expression levels of the above genes in $SOST^{ki}$ mice with disordered lipid metabolism than WT mice. Additionally, we have also evaluated the effect of Apc001OA on glucose metabolism. Of note, it was found that $SOST^{ki}$ + Apc001OA group exhibited an improvement in both glucose tolerance and insulin sensitivity, and the FBG level was significantly lower when compared to those in $SOST^{ki}$ + Veh group (Fig. 3g, h). $SOST^{ki}$ + Apc001OA group showed lower IRS1 phosphorylation but higher AKT phosphorylation in gWAT than the $SOST^{ki}$ + Veh group (Supplementary Fig 4c). Apc001OA treatment significantly lowered the expression levels of genes associated with glucose metabolism (Glut4 and IRS1) in gWAT

and iWAT of $SOST^{ki}$ mice, suggesting that Apc001OA might improve glucose metabolism through inhibiting glucose uptake in adipose tissues (Fig. 3i and Supplementary Fig 4d). The above pharmacologic evidence consistently indicated that loop3 in sclerostin played an important role in the impairment effect of sclerostin on whole-body lipid and glucose metabolism in vivo.

## Genetic mutation of loop3 attenuated the whole-body lipid and glucose metabolism disorders, as well as improved bone microarchitecture and mechanical properties in high-fat diet (HFD)-induced mice

To examine whether genetic mutation of sclerostin loop3 could attenuate HFD-induced whole-body lipid and glucose metabolism disorders in vivo, loop3 mutation ($SOST^{loop3m}$) mice were constructed, and 6-week-old male WT ($SOST^{WT}$) and $SOST^{loop3m}$ mice were fed a HFD for 16 weeks (Fig. 4a and Supplementary Fig 2c). It was found that the body weights and food intake were not changed in $SOST^{loop3m}$ + HFD group when compared to those in $SOST^{WT}$ + HFD group (Fig. 4b, c), while the fat pad weights and serum FFA levels were significantly lower in $SOST^{loop3m}$ + HFD group when compared to $SOST^{WT}$ + HFD group (Fig. 4d, e). Given that both sclerostin overproduction and sclerostin knock-out have been reported to exert no significant influence on energy intake or energy expenditure[10], the observed improvement in lipid metabolism in $SOST^{loop3m}$ + HFD group was likely attributable to the alterations in the balance between anabolic and catabolic metabolism. In accordance with these data, the sizes of individual adipocyte in gWAT and iWAT of $SOST^{loop3m}$ + HFD group were smaller than those of $SOST^{WT}$ + HFD group (Fig. 4f). Moreover, in gWAT of $SOST^{loop3m}$ + HFD group, the expression levels of genes associated with lipid anabolism were significantly lower or exhibited a lower trend (Acaca, Srebp2, Lpl and Fasn), while the expression levels of genes associated with lipid catabolism (Acads and Acadvl) were significantly higher, when compared to those of $SOST^{WT}$ + HFD group (Fig. 4g). Similarly, in iWAT of the $SOST^{loop3m}$ + HFD group, the expression levels of genes associated with lipid anabolism (Lpl and Fasn) were significantly lower, whereas the expression levels of genes associated with lipid catabolism (Acads, Cpt1a and Acadvl) were significantly higher than those in iWAT of $SOST^{WT}$ + HFD group (Supplementary Fig 5a). With the exception of the lower FBG level, $SOST^{loop3m}$ + HFD group also exhibited better glucose handling during GTT and ITT, when compared to $SOST^{WT}$ + HFD group (Fig. 4h, i). $SOST^{loop3m}$ + HFD group showed lower IRS1 phosphorylation but higher AKT phosphorylation in gWAT than the $SOST^{WT}$ + HFD group (Supplementary Fig 5b). Consistently, the expression levels of genes associated with glucose metabolism were significantly lower in both gWAT and iWAT of $SOST^{loop3m}$ + HFD group relative to those of $SOST^{WT}$ + HFD group (Fig. 4j and Supplementary Fig 5c). All the evidence indicated that genetic mutation of sclerostin loop3 attenuated whole-body lipid and glucose metabolism disorders in HFD-induced mice in vivo.

Additionally, we evaluated the effect of sclerostin loop3 mutation on bone phenotype (Supplementary Fig 6). Impressively, the data showed that the trabecular bone volume ratio (Tb.BV/TV), trabecular volumetric bone mineral density (Tb.vBMD), trabecular thickness (Tb.Th), trabecular number (Tb.N) and trabecular connectivity density (Tb.ConnD) at both the distal femur (Supplementary Fig 6a, d) and proximal tibia (Supplementary Fig 6b, e) were significantly higher in $SOST^{loop3m}$ + HFD group in comparison to those in $SOST^{WT}$ + HFD group. Meanwhile, the trabecular spacing (Tb.Sp) was significantly lower in $SOST^{loop3m}$ + HFD group than in $SOST^{WT}$ + HFD group (Supplementary Fig 6a, b, d, e). The structure model index (Tb.SMI) was close to 0 (plate-like shape of the trabecular bone) at the distal femur and proximal tibia in $SOST^{loop3m}$ + HFD group, while it was close to 3 (rod-like shape of the trabecular bone) in $SOST^{WT}$ + HFD group (Supplementary Fig 6a, b, d, e). It indicated a substantial trabecular bone improvement in $SOST^{loop3m}$ + HFD group when compared to

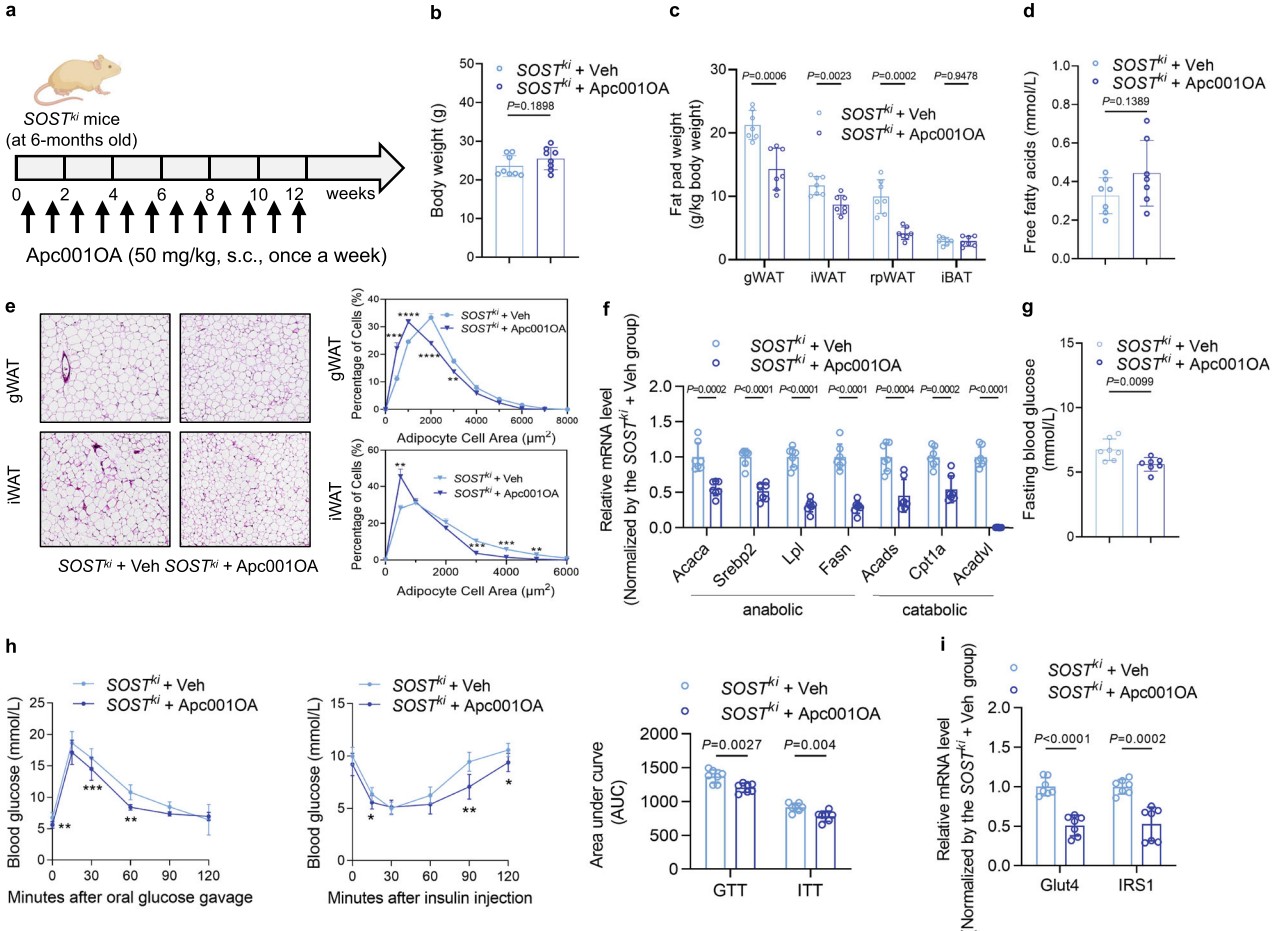

**Fig. 3 | Loop3 inhibition by the pharmacologic inhibitor Apc001OA (a sclerostin loop3-specific aptamer) attenuated the impairment effect of sclerostin on whole-body lipid and glucose metabolism in vivo. a** Experimental design. **b** Body weights of *SOST^ki^* mice with or without Apc001OA treatment. **c** Fat pad weights in *SOST^ki^* mice with or without Apc001OA treatment. gWAT: gonadal white adipose tissue; iWAT: inguinal white adipose tissue; rpWAT: retroperitoneal white adipose tissue; iBAT: interscapular brown adipose tissue. **d** Serum free fatty acids in *SOST^ki^* mice with or without Apc001OA treatment. **e** Representative images of histological sections (left) and frequency distribution of adipocyte sizes (right) in gWAT and iWAT from *SOST^ki^* mice with or without Apc001OA treatment. Scale bars, 100 μm. **f** Expression levels of genes associated with lipid anabolism (*Acaca, Srebp2, Lpl* and *Fasn*) and catabolism (*Acads, Cpt1a,* and *Acadvl*) in gWAT from *SOST^ki^* mice

with or without Apc001OA treatment detected by qPCR. **g** Fasting blood glucose in *SOST^ki^* mice with or without Apc001OA treatment. **h** Glucose tolerance test (GTT) (left), insulin tolerance test (ITT) (middle), and area under the curve (AUC) analysis for GTT and ITT (right) in *SOST^ki^* mice with or without Apc001OA treatment. **i** Expression levels of genes associated with glucose metabolism in gWAT from *SOST^ki^* mice with or without Apc001OA treatment detected by qPCR. Note: $n = 7$ biologically independent samples per group. All data were expressed as mean ± SD. *$P < 0.05$, **$P < 0.01$, ***$P < 0.001$ and ****$P < 0.0001$ for intergroup comparison. Statistical significance was calculated using unpaired *t*-test. ns no significance. All tests were two-sided. Mouse image was created in BioRender. Zhang, G. (2025) https://BioRender.com/17da8dl.

---

*SOST^WT^* + HFD group. For the femoral mid-shaft, the data showed that *SOST^loop3m^* + HFD group had significantly higher cortical polar moment of inertia (Ct.pMOI), cortical area (Ct.Ar), and total cross-sectional area (Tt.Ar) compared to *SOST^WT^* + HFD group (Supplementary Fig 6c, f). As expected, the three-point bending test data showed that stiffness, failure force, and fracture energy were significantly higher in *SOST^loop3m^* + HFD group in comparison with those in *SOST^WT^* + HFD group (Supplementary Fig 6g). The above data indicated that genetic mutation of sclerostin loop3 enhanced bone mass, improved bone microarchitecture and mechanical properties in HFD-induced mice in vivo.

### Loop3 inhibition by the pharmacologic tool Apc001OA attenuated the whole-body lipid and glucose metabolism disorders, as well as improved bone microarchitecture and mechanical properties in HFD-induced mice

To examine whether inhibition of sclerostin loop3 by the pharmacologic tool Apc001OA could attenuate HFD-induced whole-body lipid and glucose metabolism disorders in vivo, 10-week-old male C57BL/6J

mice were fed a HFD for 12 weeks. The mice were euthanized before treatment as HFD-Baseline group, and the remaining mice were treated with Vehicle (HFD + Vehicle), Apc001OA (HFD + Apc001OA), and semaglutide (HFD + Semaglutide) for 8 weeks, respectively (Semaglutide, a first-line drug for T2DM[25–27], was used as a positive control) (Fig. 5a). After HFD induction, the serum sclerostin not dickkopf-1 (DKK1) level was significantly elevated from baseline in Vehicle group. There were no differences in the serum sclerostin and DKK1 levels between HFD + Apc001OA group and HFD + Vehicle group, while HFD + Semaglutide group showed higher sclerostin level but not DKK1 level when compared to HFD + Vehicle group. No differences were found in the mRNA levels of *SOST* and *DKK1* from gWAT or iWAT (Fig. 5b, c and Supplementary Fig 7a, b). After 8 weeks of treatment, the body weights and weight of gWAT were significantly lower in both HFD + Apc001OA group and HFD + Semaglutide group than those in HFD + Vehicle group (Fig. 5d, e). Notably, the food intake was not influenced in HFD + Apc001OA group or HFD + Semaglutide group compared to HFD + Vehicle group, suggesting that Apc001OA exerts no effects on energy intake (Fig. 5f). Furthermore, the serum FFA levels

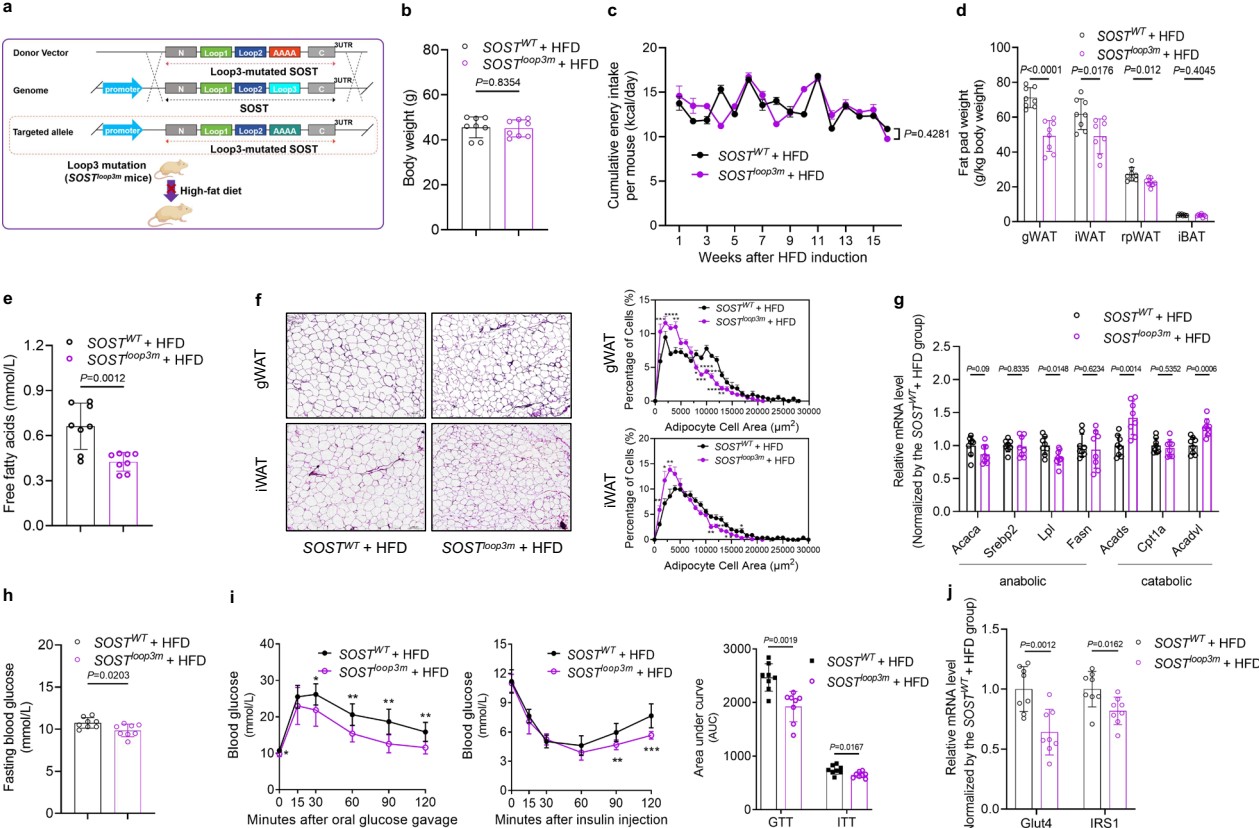

**Fig. 4 | Genetic mutation of sclerostin loop3 attenuated the whole-body lipid and glucose metabolism disorders in high-fat diet (HFD)-induced mice. a** The schematic diagram of construction strategy and experimental design. **b** Body weights of *SOST^loop3m* + HFD and *SOST^WT* + HFD mice. **c** Food intake in *SOST^loop3m* + HFD and *SOST^WT* + HFD mice. Two-way ANOVA with Sidak's multiple comparisons tests were conducted to evaluate intergroup variations. **d** Fat pad weights in *SOST^loop3m* + HFD and *SOST^WT* + HFD mice. gWAT: gonadal white adipose tissue; iWAT: inguinal white adipose tissue; rpWAT: retroperitoneal white adipose tissue; iBAT: interscapular brown adipose tissue. **e** Serum free fatty acids in *SOST^loop3m* + HFD and *SOST^WT* + HFD mice. **f** Representative images of histological sections (left) and frequency distribution of adipocyte sizes (right) in gWAT and iWAT from *SOST^loop3m* + HFD and *SOST^WT* + HFD mice. Scale bars, 200 μm.

**g** Expression levels of genes associated with lipid anabolism (*Acaca, Srebp2, Lpl* and *Fasn*) and catabolism (*Acads, Cpt1a,*and *Acadvl*) in gWAT from *SOST^loop3m* + HFD and *SOST^WT* + HFD mice detected by qPCR. **h** Fasting blood glucose in *SOST^loop3m* + HFD and *SOST^WT* + HFD mice. **i** Glucose tolerance test (GTT) (left), insulin tolerance test (ITT) (middle), and area under the curve (AUC) analysis for GTT and ITT (right) in *SOST^loop3m* + HFD and *SOST^WT* + HFD mice. **j** Expression levels of genes associated with glucose metabolism in gWAT from *SOST^loop3m* + HFD and *SOST^WT* + HFD mice detected by qPCR. Note: n = 8 biologically independent samples per group. All data were expressed as mean ± SD. *P < 0.05, **P < 0.01, ***P < 0.001 and ****P < 0.0001 for intergroup comparison. Statistical significance was calculated using unpaired *t*-test. ns no significance. All tests were two-sided. Mouse image was created in BioRender. Zhang, G. (2025) https://BioRender.com/17da8dl.

were significantly lower in HFD + Apc001OA group but higher in HFD + Semaglutide group relative to those in HFD + Vehicle group (Fig. 5g). The adipocytes in both gWAT and iWAT were smaller in HFD + Apc001OA group and HFD + Semaglutide group when compared to HFD + Vehicle group (Fig. 5h). Different from the data in *SOST^ki* mice with/without Apc001OA treatment, the expression levels of genes associated with lipid anabolism and lipid catabolism in both gWAT and iWAT were significantly higher in HFD + Apc001OA group and HFD + Semaglutide group than those in HFD + Vehicle group (Fig. 5i and Supplementary Fig 7c). Apc001OA and semaglutide might improve lipid metabolism by increasing lipid turnover rate in HFD-induced mice. In line with the data in *SOST^ki* mice with/without Apc001OA treatment, HFD + Apc001OA group and HFD + Semaglutide group exhibited better glucose handling during GTT and ITT relative to HFD + Vehicle group, but their FBG levels were comparable (Fig. 5j, k). Both HFD + Apc001OA group and HFD + Semaglutide group showed lower IRS1 phosphorylation but higher AKT phosphorylation in gWAT than the HFD + Vehicle group (Supplementary Fig 7d). Interestingly, HFD + Apc001OA group also showed lower IRS1 phosphorylation but higher AKT phosphorylation in skeletal muscle than the HFD + Vehicle group (Supplementary Fig 7e). Consistently, HFD + Apc001OA group, as well as HFD + Semaglutide group, showed significantly higher expression

levels of *Glut4* and *IRS1* in both gWAT and iWAT, when compared to HFD + Vehicle group (Fig. 5l and Supplementary Fig 7f). The above findings indicated that inhibition of sclerostin loop3 by the pharmacologic tool Apc001OA attenuated HFD-induced whole-body lipid and glucose metabolism disorders in vivo.

Thereafter, as inspired by the improving effect of loop3 mutation on bone mass, bone microarchitecture, and mechanical properties in HFD-induced mice, we also evaluated the effect of loop3 inhibition by the pharmacologic tool Apc001OA on bone phenotype (Supplementary Fig 8). The data showed that the Tb.BV/TV, Tb.vBMD, Tb.Th, Tb.N and Tb.ConnD at both the distal femur (Supplementary Fig 8a, d) and proximal tibia (Supplementary Fig 8b, e) were significantly higher in HFD + Apc001OA group in comparison to those in HFD + Vehicle group. Additionally, the Tb.Sp was significantly lower in HFD + Apc001OA group than HFD + Vehicle group. The Tb.SMI was closer to 0 at the distal femur and proximal tibia in HFD + Apc001OA group when compared to HFD + Vehicle group (Supplementary Fig 8a, b, d, e). However, for the distal femur, the HFD + Semaglutide group showed lower Tb.BV/TV, Tb.N and Tb.ConnD but higher Tb.Sp than HFD + Vehicle group. For the proximal tibia, the HFD + Semaglutide group showed lower Tb.N and Tb.ConnD but higher Tb.Sp than HFD + Vehicle group (Supplementary Fig 8a, b, d, e). For the femoral mid-

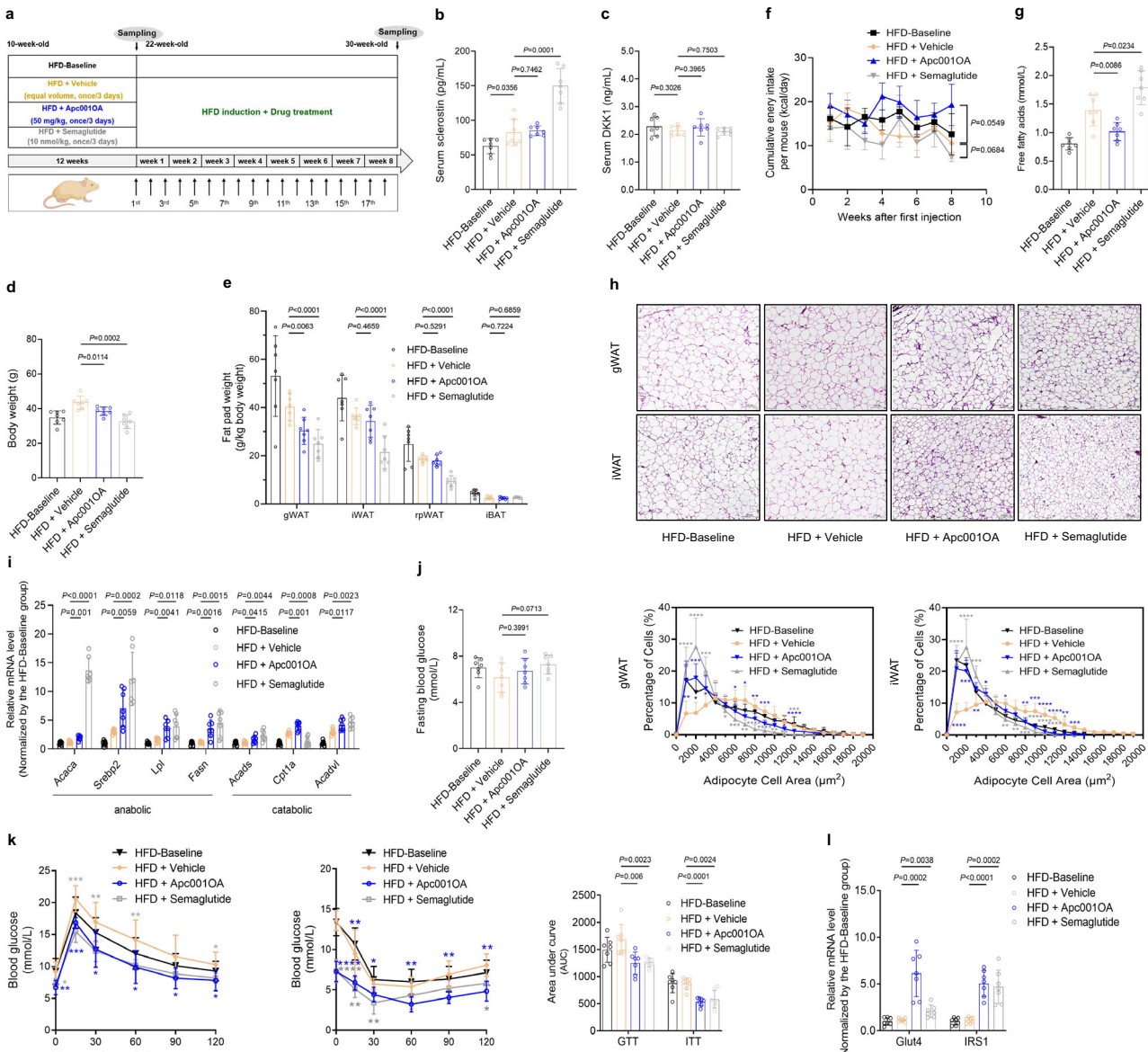

**Fig. 5 | Loop3 inhibition by the pharmacologic inhibitor Apc001OA (a sclerostin loop3-specific aptamer) ameliorated high-fat diet (HFD)-induced lipid and glucose metabolism disorders. a** Experimental design. **b** Body weights of HFD-induced mice with or without Apc001OA treatment. **c** Serum sclerostin levels in HFD-induced mice with or without Apc001OA treatment. **d** Serum DKK1 levels in HFD-induced mice with or without Apc001OA treatment. **e** Fat pad weights in HFD-induced mice with or without Apc001OA treatment. gWAT: gonadal white adipose tissue; iWAT: inguinal white adipose tissue; rpWAT: retroperitoneal white adipose tissue; iBAT: interscapular brown adipose tissue. **f** Food intake in HFD-induced mice with or without Apc001OA treatment. Two-way ANOVA with Sidak's multiple comparisons test was conducted to evaluate intergroup variations. **g** Serum free fatty acids in HFD-induced mice with or without Apc001OA treatment. **h** Representative images of histological sections (left) and frequency distribution of adipocyte sizes (right) in gWAT and iWAT from HFD-induced mice with or without Apc001OA treatment. Scale bars, 200 μm. **i** Expression levels of genes associated

with lipid anabolism (*Acaca, Serbp2, Lpl,* and *Fasn*) and catabolism (*Acads, Cpt1a,* and *Acadvl*) in gWAT from HFD-induced mice with or without Apc001OA treatment detected by qPCR. **j** Fasting blood glucose in HFD-induced mice with or without Apc001OA treatment. **k** Glucose metabolism in HFD-induced mice with or without Apc001OA treatment. Glucose tolerance test (GTT) (left), insulin tolerance test (ITT) (middle), and area under the curve (AUC) analysis for GTT and ITT (right). **l** Expression levels of genes associated with glucose metabolism in gWAT from HFD-induced mice with or without Apc001OA treatment detected by qPCR. Note: $n = 7$ biologically independent samples per group. All data were expressed as mean ± SD. *$P < 0.05$, **$P < 0.01$, ***$P < 0.001$ and ****$P < 0.0001$ for intergroup comparison (blue *: HFD + Vehicle vs. HFD + Apc001OA; gray *: HFD + Vehicle vs. HFD + Semaglutide). Statistical significance was calculated using unpaired *t*-test. ns: no significance. All tests were two-sided. Mouse image was created in BioRender. Zhang, G. (2025) https://BioRender.com/17da8dl.

shaft, the data showed that HFD + Apc001OA group had significantly higher Ct.pMOI, Ct.Ar, and Tt.Ar compared to HFD + Vehicle group, while no difference was observed between HFD + Semaglutide group and HFD + Vehicle group (Supplementary Fig 8c,f). Further, the three-point bending test data showed that stiffness, failure force and fracture energy were significantly higher in HFD + Apc001OA group compared to those in HFD + Vehicle group (Supplementary Fig 8g).

Comparatively, there were no obvious differences in the above parameters between HFD + Semaglutide group and HFD + Vehicle group (Supplementary Fig 8g). Collectively, pharmacologic inhibition of sclerostin loop3 by Apc001OA enhanced bone mass, improved bone microarchitecture and mechanical properties in HFD-induced mice in vivo.

## Adipocytic sclerostin loop3-LRP4 interaction was required by sclerostin to impair lipid and glucose metabolism in vitro

Based on the above findings, sclerostin loop3 was revealed to participate in the impairment effect of sclerostin on whole-body lipid and glucose metabolism, and targeting sclerostin loop3 could attenuate the HFD-induced whole-body lipid and glucose metabolism disorders. However, it is unclear how sclerostin loop3 participates in the impairment effect of sclerostin on whole-body lipid and glucose metabolism. LRP4 is a member of the LDL receptor family, which consists of 8 LDL-receptor class A (LA) domains, 20 LDL-receptor class B (LB) domains, 3 EGF-like domains, and some disordered domains[28]. It has been established that LRP4 is required for the endocrine function of sclerostin[29]. Therefore, it is of great interest to investigate whether sclerostin loop3 could interact with LRP4 to mediate sclerostin-impaired lipid and glucose metabolism in vitro.

In this study, the pull-down assay was first utilized to demonstrate the binding between sclerostin/sclerostin loop3 and LRP4. Consistently, it was found that both sclerostin and sclerostin loop3 bound to LRP4, while sclerostin loop2 did not bind with LRP4 (Fig. 6a, Supplementary Fig. 9a). Apc001 targeting loop3 could block the binding of sclerostin to LRP4 by biolayer interferometry (BLI). The $K_d$ of sclerostin and sclerostin loop3 to LRP4 was 1.98 nM and 6.0 nM, respectively, which was in accordance with previously reported binding affinity ($K_d = 0.73$ nM) between sclerostin and LRP4 (Fig. 6b–d)[30]. The sclerostin loop3 did not bind with LRP6 (Supplementary Fig. 9b). Then, *Lrp4* plasmids encoding full-length LRP4 and LRP4 truncations were constructed to determine the interactive domains between sclerostin loop3 and LRP4 in vitro (Supplementary Table 1). After protein over-expression for pull-down assay, it was found that the binding between sclerostin loop3 and LRP4-T1/T2/T3/T4/T5/T6 truncations still existed, whereas there was no binding between sclerostin loop3 and LRP4-T6_1/T6_2/T6_3 truncations, demonstrating that the domain (LA5) removed from C-terminus of LRP4-T6 was critical for the binding of LRP4 to sclerostin loop3 (Supplementary Fig 9c). To further identify residues responsible for the binding of LRP4 to sclerostin loop3, twelve three-site directed mutations were introduced to *Lrp4* plasmid. In pull-down assay, the FLAG-LRP4 bands intensity was lower in LRP4-m4, LRP4-m6, and LRP4-m7 groups, when compared to LRP4 group (Supplementary Fig 9d). It suggested that the residues Y200, G201, L205, D206, I207, Y208, H209, and C210 within the above three muteins could be the key binding sites of LRP4 to sclerostin loop3. By combining the mutations in LRP4-m4, LRP4-m6 and LRP4-m7 muteins, it was found that muteins LRP4-m46, LRP4-m47 and LRP4-67 showed weaker binding to sclerostin loop3 (Fig. 6e). Subsequently, the Wnt signaling in sclerostin knock-out 3T3-L1 cells with over-expression of *Lrp4m* (encoding LRP4-m47 mutein with Y200A, G201A, Y208A, H209A and C210A) was partially recovered in the presence of sclerostin compared to that in sclerostin knock-out 3T3-L1 cells with over-expression of *Lrp4*, while *Lrp4* mutation could not change the Wnt signaling in the absence of sclerostin (Fig. 6f). It demonstrated that mutation of key binding sites (Y200, G201, Y208, H209 and C210) within LRP4 to sclerostin loop3 reactivated Wnt signaling antagonized by sclerostin through specific blockade of sclerostin loop3-LRP4 interaction in vitro, which might exert functions on lipid and glucose metabolism.

To genetically investigate whether sclerostin loop3-LRP4 interaction participated in the impairment effect of sclerostin on lipid and glucose metabolism in vitro, sclerostin knock-out 3T3-L1 cells with over-expression of LRP4 protein (*Lrp4* group) and LRP4-m47 mutein (*Lrp4m* group) were treated with recombinant sclerostin protein (mScl). The Oil red O staining data showed that the lipid droplet formation in *Lrp4* group was greater than that in *Lrp4m* group in the presence of sclerostin during adipogenic induction, whereas there was no difference in the lipid droplet formation between *Lrp4* group and

*Lrp4m* group in the absence of sclerostin (Fig. 6g). Additionally, in *Lrp4* + mScl group, the expression levels of lipid anabolism and catabolism markers were higher than those in *Lrp4m* + mScl group, while there were no differences between *Lrp4* group and *Lrp4*m group (Fig. 6h, i). Similarly, the expression level of *Glut4* was significantly lower in *Lrp4m* + mScl group than that in *Lrp4* + mScl group, but the expression level of *IRS1* was not influenced. No difference was observed in the expression levels of *Glut4* and *IRS1* in *Lrp4* group and *Lrp4*m group in vitro (Fig. 6j). Not surprisingly, both the glucose uptake and insulin-stimulated glucose uptake were significantly higher in *Lrp4* + mScl group compared to those in *Lrp4m* + mScl group, while they were comparable in *Lrp4* group and *Lrp4m* group (Fig. 6k, l). Therefore, we identified mutations of key binding sites (Y200, G201, Y208, H209, and C210) within LRP4 that could decrease the binding between sclerostin loop3 and LRP4 but did not influence the lipid and glucose metabolism in the absence of sclerostin in vitro. Collectively, the above data showed that *Lrp4* mutation-induced blockade of sclerostin loop3-LRP4 interaction attenuated the impairment effect of sclerostin on lipid and glucose metabolism in 3T3-L1 cells in vitro.

To pharmacologically investigate whether sclerostin loop3-LRP4 interaction participated in the impairment effect of sclerostin on whole-body lipid and glucose metabolism in vitro, we designed and synthesized a blocking peptide LA5 based on the pull-down assay. BLI analysis validated that LA5 could bind to sclerostin ($K_d = 11.3$ nM) and sclerostin loop3 ($K_d = 155$ nM) (Supplementary Fig 9e, f). The pre-incubation of sclerostin with LA5 could block the sclerostin-LRP4 interaction (Supplementary Fig 9g). Further, it was found that LA5 treatment partially recovered the Wnt signaling antagonized by sclerostin in sclerostin knock-out 3T3-L1 cells with over-expression of LRP4 in the presence of sclerostin, while LA5 could not change the Wnt signaling in sclerostin knock-out 3T3-L1 cells with over-expression of LRP4 in the absence of sclerostin (Fig. 7a). Importantly, the Oil red O staining data showed that the lipid droplet formation in *Lrp4* + mScl group was greater than that in *Lrp4* group, demonstrating the promoting effect of sclerostin on lipid droplet formation. There was no noticeable difference in the lipid droplet formation between *Lrp4* group and *Lrp4* + LA5 group, which demonstrated that LA5 itself did not affect the lipid droplet formation in the absence of sclerostin. The lipid droplet formation promoted by sclerostin-LRP4 interaction in *Lrp4* + mScl group was significantly inhibited by LA5 in *Lrp4* + mScl + LA5 group (Fig. 7b). In *Lrp4* + mScl + LA5 group, the expression levels of lipid anabolism and catabolism markers were significantly lower than those in *Lrp4* + mScl group, whereas the LA5 treatment made no difference in the absence of sclerostin in vitro (Fig. 7c, d). Similarly, the expression level of *Glut4* was significantly lower in *Lrp4* + mScl + LA5 group than *Lrp4* + mScl group, albeit that the expression level of *IRS1* was not changed, suggesting that LA5 might not function through the downregulation of mRNA level of *IRS1*. No difference was observed in the expression level of *Glut4* or *IRS1* between *Lrp4* group and *Lrp4* + LA5 group in vitro (Fig. 7e). Furthermore, the glucose uptake and insulin-stimulated glucose uptake promoted by sclerostin in *Lrp4* + mScl group were attenuated in *Lrp4* + mScl + LA5 group. The LA5 itself could not influence the glucose uptake in the absence of sclerostin, which was suggestive of that LA5 exerted functions in the regulation of glucose metabolism by blocking sclerostin loop3-LRP4 interaction (Fig. 7f, g). Therefore, we here designed a blocking peptide LA5, that could decrease the binding between sclerostin loop3 and LRP4 but did not influence the lipid and glucose metabolism in the absence of sclerostin in vitro. The above data showed that the blocking peptide LA5-induced blockade of sclerostin loop3-LRP4 interaction attenuated the impairment effect of sclerostin on lipid and glucose metabolism in 3T3-L1 cells in vitro.

Taken together, both our genetic and pharmacologic evidence indicated that adipocytic sclerostin loop3-LRP4 interaction was required by sclerostin to impair lipid and glucose metabolism in vitro.

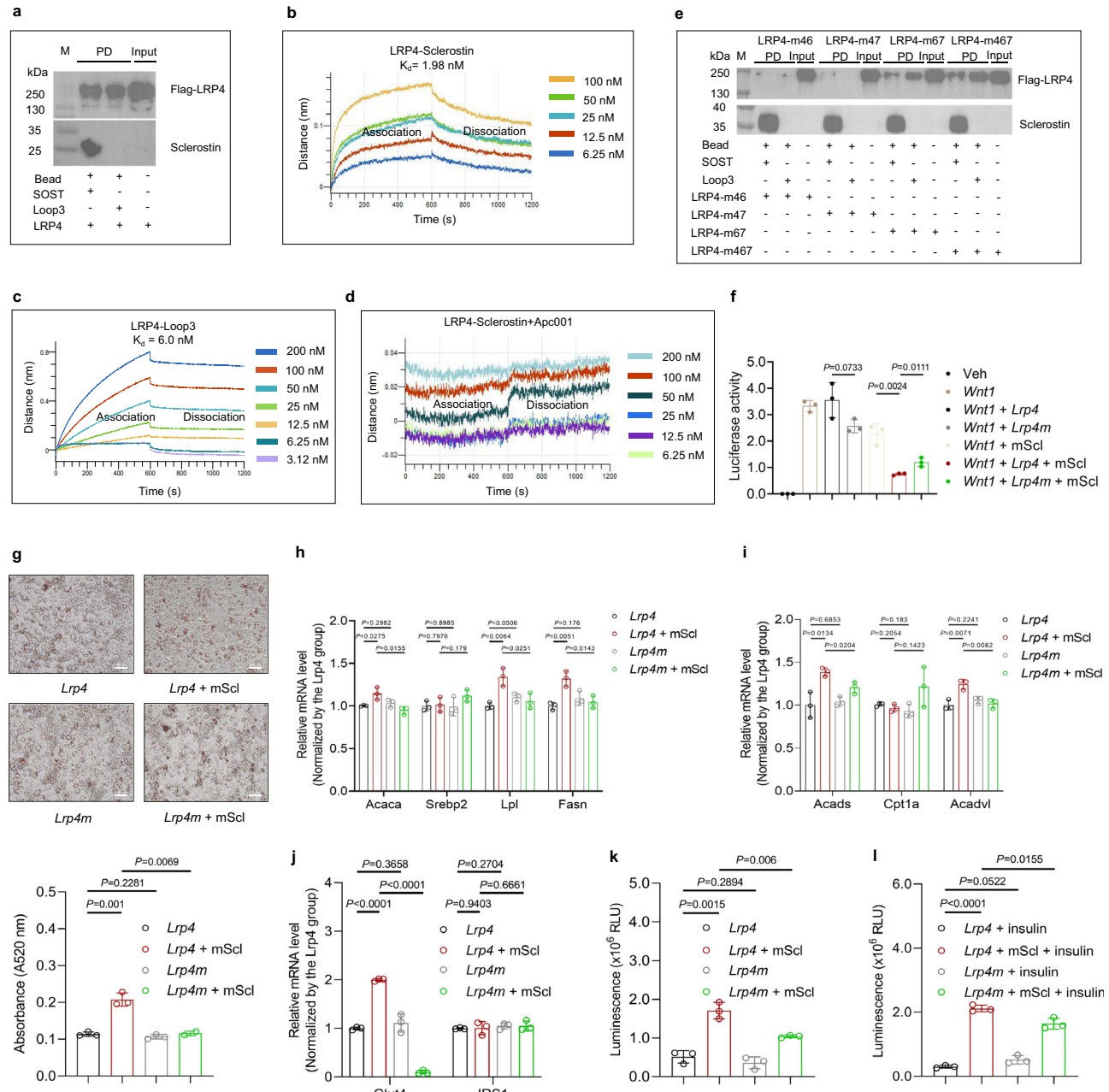

**Fig. 6 | The impairment effect of sclerostin on lipid and glucose metabolism in 3T3-L1 cells (pre-adipocytes) was attenuated upon LRP4 mutation (Y200A, G201A, Y208A, H209A, C210A) in vitro. a** Binding ability of sclerostin to LRP4 by pull-down assay. **b** BLI analysis of the binding affinity between sclerostin and LRP4. (**c**) BLI analysis of the binding affinity between sclerostin loop3 and LRP4. **d** BLI analysis of the binding affinity between sclerostin and LRP4 in the presence of Apc001. **e** Binding ability of LRP4 muteins (LRP4 m46, m47, m67, m467) to sclerostin loop3 by pull-down assay for identifying the binding residues on LRP4 to sclerostin loop3. **f** TOP-Wnt luciferase signaling in 3T3-L1 cells with expression of LRP4 or LRP4 mutein in vitro in the presence of sclerostin. **g** Lipid droplet formation staining (upper) and quantification (lower) in 3T3-L1 cells with expression of LRP4 or LRP4 mutein in vitro in the presence of sclerostin. Scale bars, 100 μm.

**h** qPCR analysis of lipid anabolism markers in 3T3-L1 cells with expression of LRP4 or LRP4 mutein in vitro in the presence of sclerostin. (**i**) qPCR analysis of lipid catabolism markers in 3T3-L1 cells with expression of LRP4 or LRP4 mutein in vitro in the presence of sclerostin. (**j**) qPCR analysis of glucose metabolism markers in 3T3-L1 cells with expression of LRP4 or LRP4 mutein in vitro in the presence of sclerostin. **k** Glucose uptake in 3T3-L1 cells with expression of LRP4 or LRP4 mutein in vitro in the presence of sclerostin. **l** Insulin-stimulated glucose uptake in 3T3-L1 cells with expression of LRP4 or LRP4 mutein in vitro in the presence of sclerostin. Note: 3T3-L1 cells: pre-adipocytes. PD: pull-down. $n = 3$ biologically independent samples. All data were expressed as mean ± SD. Statistical significance was calculated using unpaired $t$-test. ns no significance. All tests were two-sided.

## Adipocytic sclerostin loop3-LRP4 interaction was required by sclerostin to impair whole-body lipid and glucose metabolism in vivo

To examine whether adipocytic sclerostin loop3-LRP4 interaction was required by sclerostin to impair whole-body lipid and glucose metabolism in vivo, global *Lrp4* mutation (*Lrp4m*) mice (*Lrp4m*, encoding LRP4-Y200A&G201A&Y208A&H209A&C210A), sclerostin knock-out (*SOST*[ko]) mice, and sclerostin knock-in (*SOST*[ki]) mice were engaged (Supplementary Fig 2d–g).

First, we established the *Lrp4m* mice and mice with global *Lrp4* (except *Lrp4* in adipocytes) mutation wherein the mutated *Lrp4* gene in adipocytes was corrected (hereafter *Lrp4m/Ad-Lrp4* mice) by the

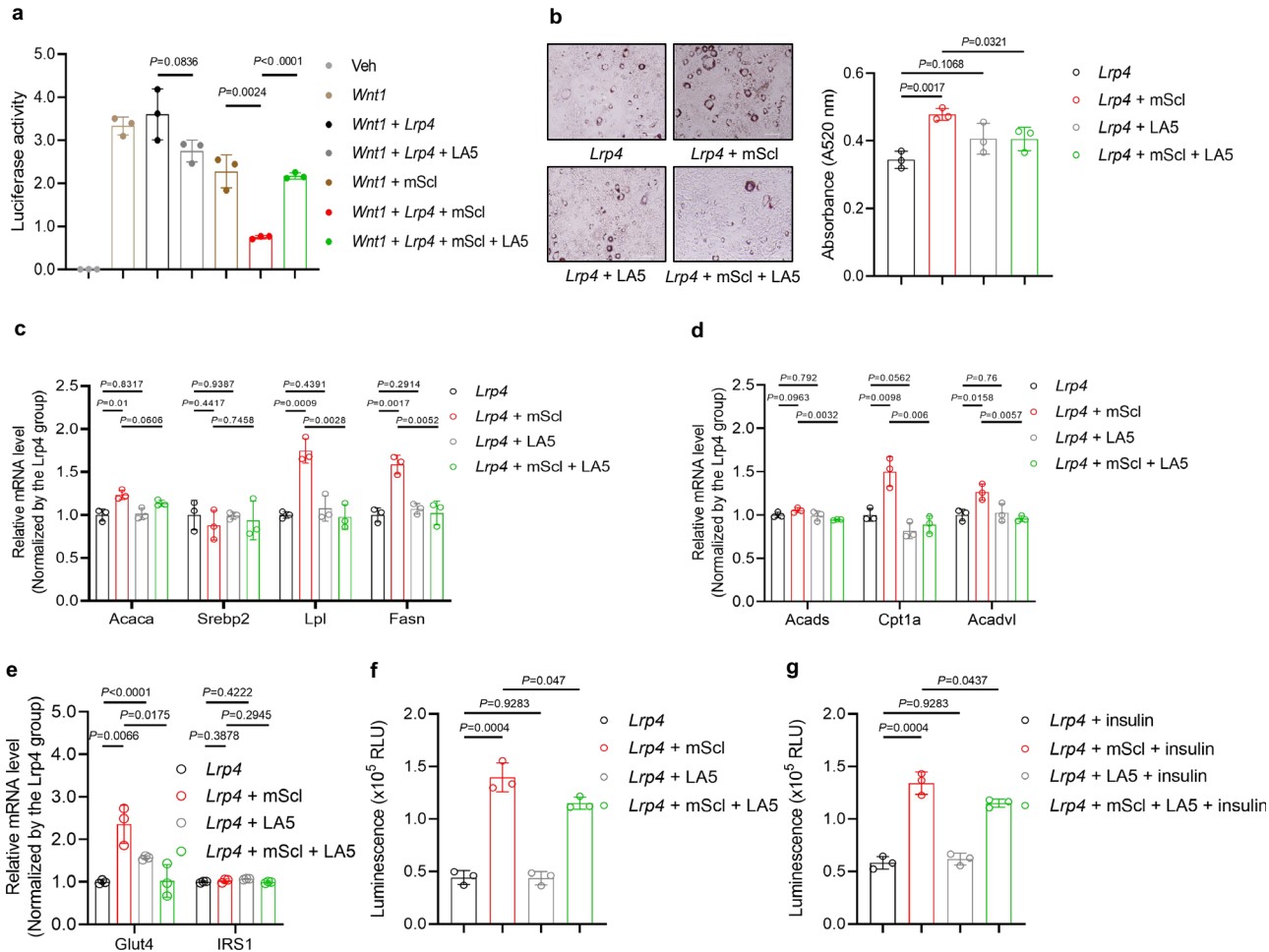

**Fig. 7 | Blockade of sclerostin loop3-LRP4 interaction by blocking peptide LA5 attenuated the impairment effect of sclerostin on lipid and glucose metabolism in 3T3-L1 cells (pre-adipocytes). a** TOP-Wnt luciferase signaling in LRP4-overexpressing 3T3-L1 cells with LA5 treatment in vitro in the presence of sclerostin. **b** Lipid droplet formation staining (left) and quantification (right) in LRP4-overexpressing 3T3-L1 cells with LA5 treatment in vitro in the presence of sclerostin. Scale bars, 100 μm. (**c**) qPCR analysis of lipid anabolism markers in LRP4-overexpressing 3T3-L1 cells with LA5 treatment in vitro in the presence of sclerostin. **d** qPCR analysis of lipid catabolism markers in LRP4-overexpressing 3T3-L1 cells with LA5 treatment in vitro in the presence of sclerostin. **e** qPCR analysis of glucose metabolism markers in LRP4-overexpressing 3T3-L1 cells with LA5 treatment in vitro in the presence of sclerostin. **f** Glucose uptake in LRP4-overexpressing 3T3-L1 cells with LA5 treatment in vitro in the presence of sclerostin. **g** Insulin-stimulated glucose uptake in LRP4-overexpressing 3T3-L1 cells with LA5 treatment in vitro in the presence of sclerostin. Note: 3T3-L1 cells: pre-adipocytes. $n = 3$ biologically independent samples. All data were expressed as mean ± SD. Statistical significance was calculated using unpaired $t$-test. ns no significance. All tests were two-sided.

adiponectin promoter-driven Cre recombination (Adipoq-Cre) (Fig. 8a and Supplementary Fig 10). The differences in lipid and glucose metabolism among 6-month-old WT mice, *Lrp4m* mice, and *Lrp4m/Ad-Lrp4* mice were compared. The data from WT mice and *Lrp4m* mice showed that *Lrp4* mutation improved the whole-body lipid and glucose metabolism in vivo (Fig. 8 and Supplementary Fig 11). Meanwhile, in *Lrp4m/Ad-Lrp4* mice, fat pad weights and serum FFA levels were higher than those in *Lrp4m* mice, whereas the body weights remained similar to those of *Lrp4m* mice (Fig. 8b–d). On the other hand, histological analyses showed that the adipocyte sizes in both gWAT and iWAT of *Lrp4m/Ad-Lrp4* mice were larger than those of *Lrp4m mice* (Fig. 8e). The expression levels of genes associated with lipid anabolism (*Acaca, Srebp2, Lpl*, and *Fasn*) were higher, but the expression levels of genes associated with lipid catabolism (*Acads, Cpt1a*, and *Acadvl*) were lower in both gWAT and iWAT of *Lrp4m/Ad-Lrp4* mice, when compared to those of *Lrp4m* mice (Fig. 8f and Supplementary Fig 11a). Although no difference was observed in FBG level between *Lrp4m* mice and *Lrp4m/Ad-Lrp4* mice, *Lrp4m/Ad-Lrp4* mice exhibited impaired glucose tolerance and insulin sensitivity relative to *Lrp4m* mice (Fig. 8g, h). In fact, LRP4 deficiency in adipocytes also showed no effect on blood glucose

reportedly[29]. We observed a lower phosphorylation level of IRS1 and a higher phosphorylation level of AKT in gWAT of *Lrp4m* mice compared to WT mice, while *Lrp4m/Ad-Lrp4* mice showed a higher phosphorylation level of IRS1 and a lower phosphorylation level of AKT than *Lrp4m* mice (Supplementary Fig 11b). The expression levels of *Glut4* and *IRS1* in both gWAT and iWAT of *Lrp4m/Ad-Lrp4* mice were lower than those of *Lrp4m* mice (Fig. 8i and Supplementary Fig 11c). The above evidence indicated that adipocytic *Lrp4m* mutation improved whole-body lipid and glucose metabolism in vivo.

Second, to determine whether the improving effect of adipocytic *Lrp4m* mutation on whole-body lipid and glucose metabolism was induced by genetic blockade of sclerostin loop3-LRP4 interaction in vivo, the distinctive responses of 6-month-old *Lrp4m/SOST^{ko}* mice and *SOST^{ko}* mice to sclerostin overproduction by injection of AAV-SOST in whole-body lipid and glucose metabolism were examined (Fig. 9a). The serum sclerostin levels were significantly higher in *Lrp4m/SOST^{ko}* mice and *SOST^{ko}* mice with sclerostin overproduction compared to corresponding mice without sclerostin overproduction, while the body weights remained almost unchanged (Fig. 9b, Supplementary Fig 12a). Consistently, it was found that *SOST^{ko}* + SOST group

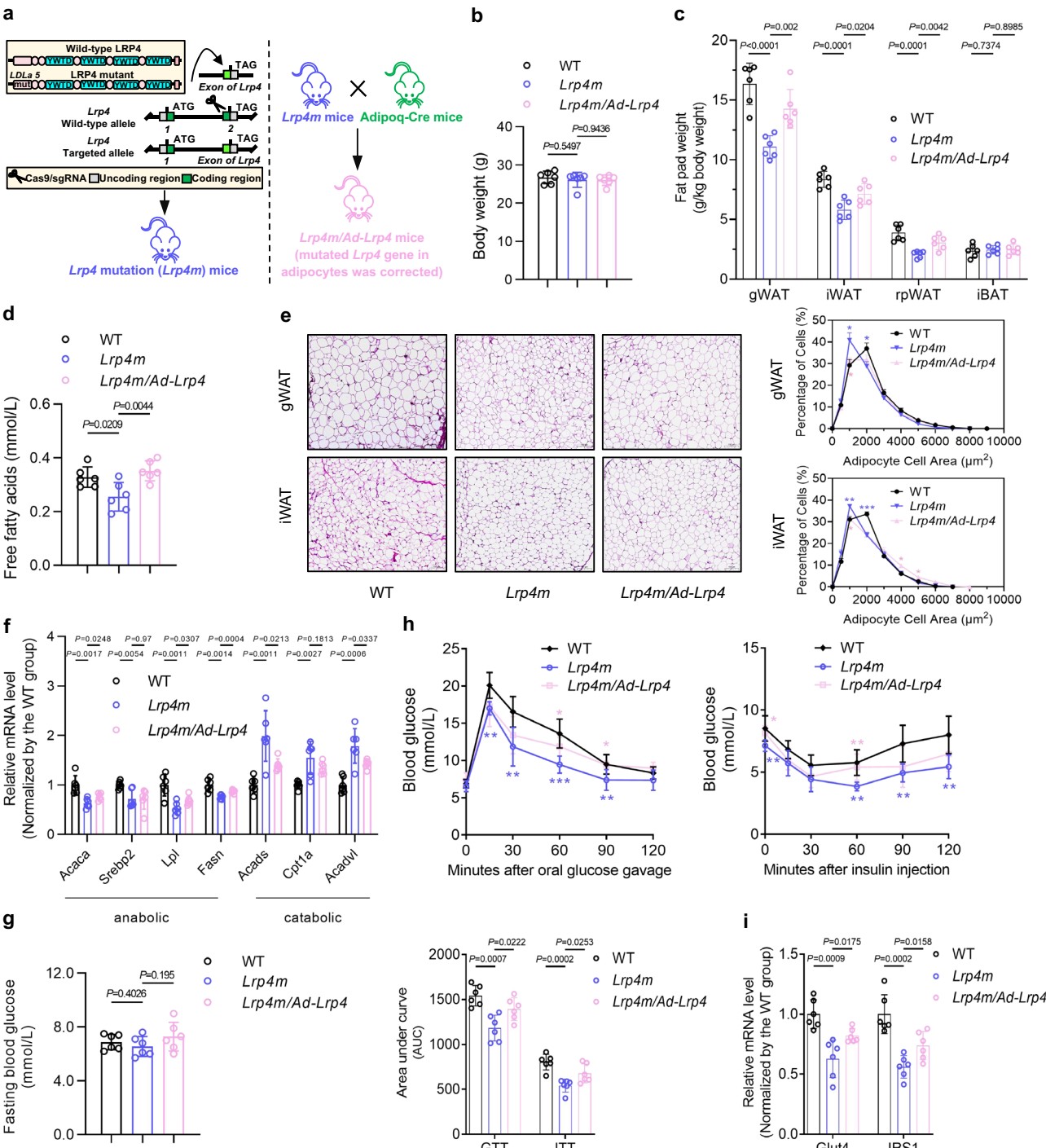

**Fig. 8 | Correction of *Lrp4* mutation to wild-type *Lrp4* in adipocytes (*Lrp4m/Ad-Lrp4*) attenuated the improvement effect of global *Lrp4* mutation (*Lrp4m*) on lipid and glucose metabolism in mice. a** The schematic diagram of construction strategy for mouse models. **b** Body weights of wild-type (WT) mice, *Lrp4m* mice, and *Lrp4m/Ad-Lrp4* mice. **c** Fat pad weights in WT mice, *Lrp4m* mice, and *Lrp4m/Ad-Lrp4* mice. gWAT: gonadal white adipose tissue; iWAT: inguinal white adipose tissue; rpWAT: retroperitoneal white adipose tissue; iBAT: interscapular brown adipose tissue. **d** Serum free fatty acids in WT mice, *Lrp4m* mice, and *Lrp4m/Ad-Lrp4* mice. **e** Representative images of histological sections (left) and frequency distribution of adipocyte sizes (right) in gWAT and iWAT from WT mice, *Lrp4m* mice, and *Lrp4m/Ad-Lrp4* mice. Scale bars, 100 μm. **f** Expression levels of genes associated with lipid

anabolism (*Acaca*, *Srebp2*, *Lpl*, and *Fasn*) and catabolism (*Acads*, *Cpt1a* and *Acadvl*) in gWAT from WT mice, *Lrp4m* mice, and *Lrp4m/Ad-Lrp4* mice detected by qPCR.
**g** Fasting blood glucose in WT mice, *Lrp4m* mice, and *Lrp4m/Ad-Lrp4* mice. **h** Glucose metabolism in WT mice, *Lrp4m* mice, and *Lrp4m/Ad-Lrp4* mice. Glucose tolerance test (GTT) (left), insulin tolerance test (ITT) (right), and area under the curve (AUC) analysis for GTT and ITT (lower). **i** Expression levels of genes associated with glucose metabolism in gWAT from WT mice, *Lrp4m* mice, and *Lrp4m/Ad-Lrp4* mice detected by qPCR. Note: *n* = 6 biologically independent samples per group. All data were expressed as mean ± SD. *$P < 0.05$, **$P < 0.01$, and ***$P < 0.001$ for intergroup comparison (blue *: WT vs. *Lrp4m*; pink *: *Lrp4m* vs. *Lrp4m/Ad-Lrp4*). Statistical significance was calculated using unpaired *t*-test. ns no significance. All tests were two-sided.

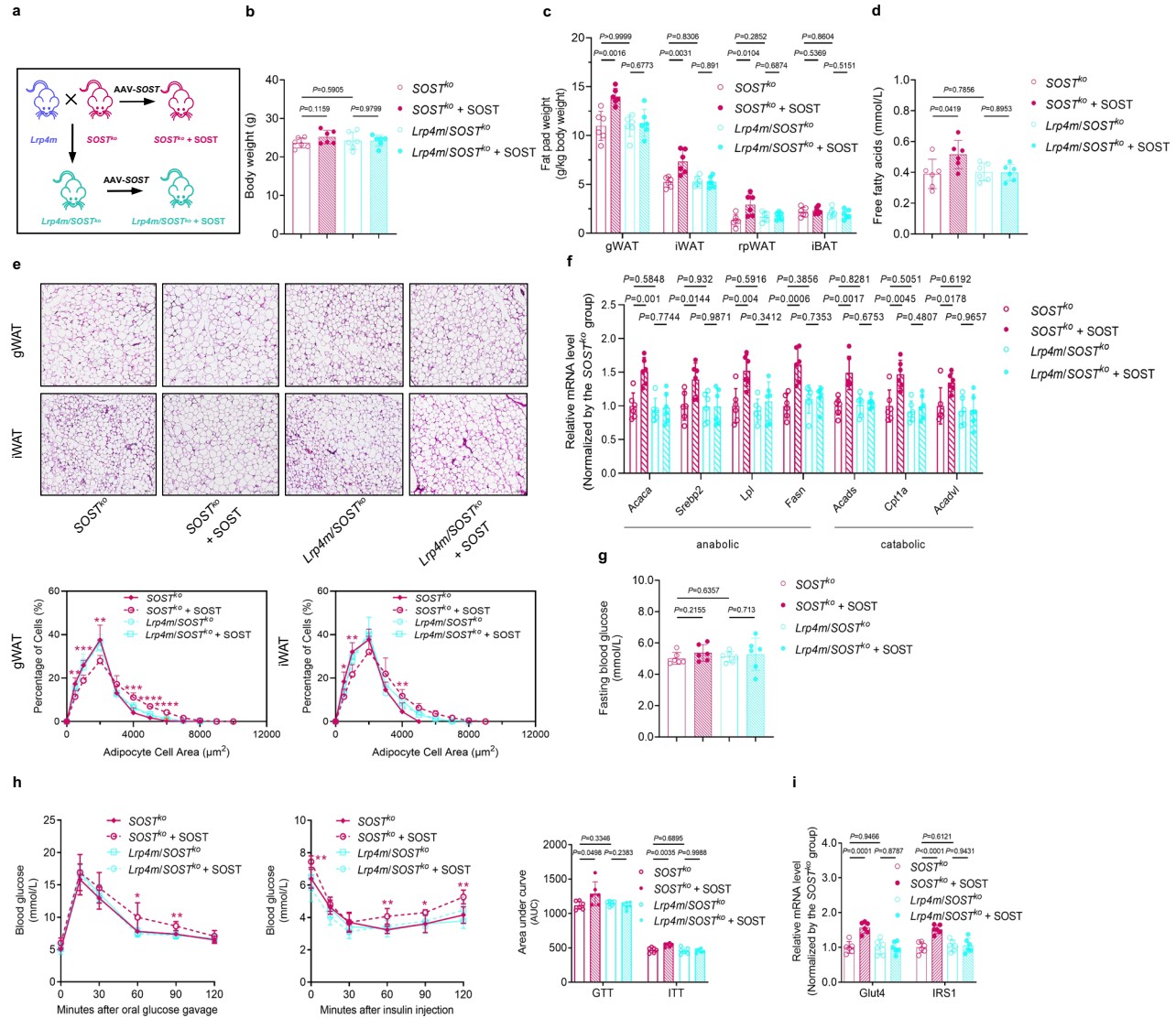

**Fig. 9 | Lipid and glucose metabolism in *Lrp4m/SOST^ko* mice and *SOST^ko* mice with or without sclerostin overproduction in vivo. a** The schematic diagram of construction strategy for mouse models. **b** Body weights of *Lrp4m/SOST^ko* mice and *SOST^ko* mice with or without sclerostin overproduction. **c** Fat pad weights in *Lrp4m/SOST^ko* mice and *SOST^ko* mice with or without sclerostin overproduction. gWAT: gonadal white adipose tissue; iWAT: inguinal white adipose tissue; rpWAT: retroperitoneal white adipose tissue; iBAT: interscapular brown adipose tissue. **d** Serum free fatty acids in *Lrp4m/SOST^ko* mice and *SOST^ko* mice with or without sclerostin overproduction. **e** Representative images of histological sections (upper) and frequency distribution of adipocyte sizes (lower) in gWAT and iWAT from *Lrp4m/SOST^ko* mice and *SOST^ko* mice with or without sclerostin overproduction. Scale bars, 100 μm. **f** Expression levels of genes associated with lipid anabolism (*Acaca*, *Srebp2*, *Lpl*, and *Fasn*) and catabolism (*Acads*, *Cpt1a*, and *Acadvl*) in gWAT from *Lrp4m/*

*SOST^ko* mice and *SOST^ko* mice with or without sclerostin overproduction detected by qPCR. **g** Fasting blood glucose in *Lrp4m/SOST^ko* mice and *SOST^ko* mice with or without sclerostin overproduction. **h** Glucose metabolism in *Lrp4m/SOST^ko* mice and *SOST^ko* mice with or without sclerostin overproduction. Glucose tolerance test (GTT) (left), insulin tolerance test (ITT) (middle), and area under the curve (AUC) analysis for GTT and ITT (right). **i** Expression levels of genes associated with glucose metabolism in gWAT from *Lrp4m/SOST^ko* mice and *SOST^ko* mice with or without sclerostin overproduction detected by qPCR. Note: *n* = 6 biologically independent samples per group. All data were expressed as mean ± SD. *$P < 0.05$, **$P < 0.01$, ***$P < 0.001$, ****$P < 0.0001$ for intergroup comparison (*SOST^ko* vs. *SOST^ko* + SOST). Statistical significance was calculated using unpaired *t*-test. ns: no significance. All tests were two-sided.

exhibited significantly higher fat pad weights and serum FFA levels than *SOST^ko* group, whereas there were no significant differences between *Lrp4m/SOST^ko* + SOST group and *Lrp4m/SOST^ko* group (Fig. 9c, d). As reported by Kim *et al.*, sclerostin overproduction and sclerostin knock-out showed no effect on body weights in mice[10]. It might be attributed to the different effects of sclerostin on fat mass and bone mass. We also noticed that adipocytes in both gWAT and iWAT of *SOST^ko* + SOST group were larger than those of *SOST^ko* group, but there were no significant differences between *Lrp4m/SOST^ko* + SOST group and *Lrp4m/SOST^ko* group (Fig. 9e). Additionally, higher expression levels of genes associated with lipid anabolism and lipid catabolism in both gWAT and iWAT of *SOST^ko* + SOST group were found when

compared to those of *SOST^ko* group, whereas there were no significant changes between *Lrp4m/SOST^ko* + SOST group and *Lrp4m/SOST^ko* group (Fig. 9f, Supplementary Fig 12b). The sclerostin overproduction did not influence the FBG level in either *SOST^ko* mice or *Lrp4m/SOST^ko* mice (Fig. 9g). The GTT and ITT data demonstrated that sclerostin overproduction impaired glucose tolerance and insulin sensitivity in *SOST^ko* mice, but *Lrp4m/SOST^ko* mice showed no response to sclerostin overproduction (Fig. 9h). Sclerostin overproduction increased the IRS1 phosphorylation but decreased the AKT phosphorylation in gWAT of *SOST^ko* mice, whereas no such effects were observed in *Lrp4m/SOST^ko* mice (Supplementary Fig 12c). As expected, higher expression levels of *Glut4* and *IRS1* in both gWAT and iWAT of *SOST^ko* mice but not

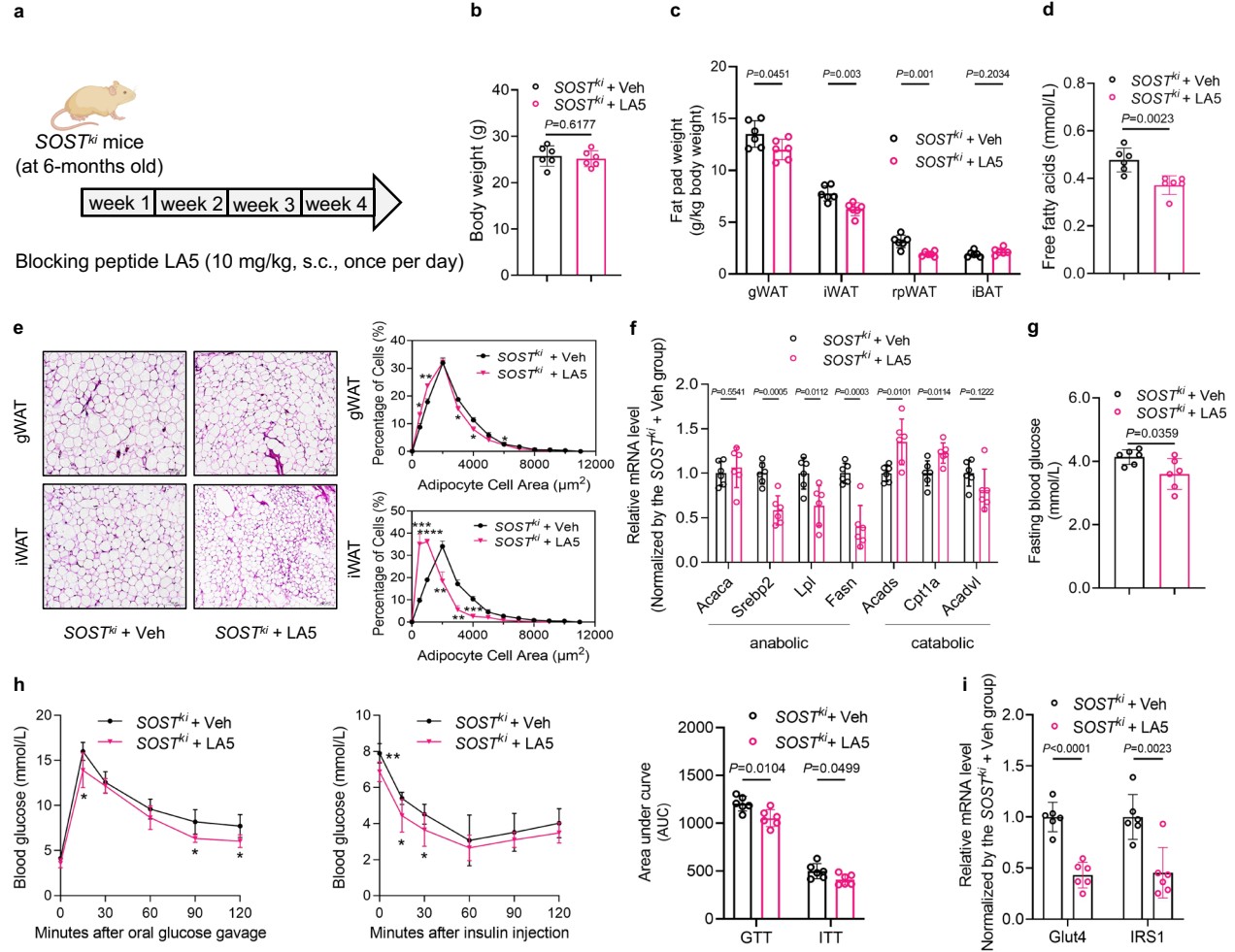

**Fig. 10 | Pharmacologic blockade of loop3-LRP4 interaction by blocking peptide LA5 improved the whole-body lipid and glucose metabolism in *SOST^ki* mice. a** Experimental design. **b** Body weights of *SOST^ki* mice with or without LA5 treatment. **c** Fat pad weights in *SOST^ki* mice with or without LA5 treatment. gWAT: gonadal white adipose tissue; iWAT: inguinal white adipose tissue; rpWAT: retroperitoneal white adipose tissue; iBAT: interscapular brown adipose tissue. **d** Serum free fatty acids in *SOST^ki* mice with or without LA5 treatment. **e** Representative images of histological sections (left) and frequency distribution of adipocyte sizes (right) in gWAT and iWAT from *SOST^ki* mice with or without LA5 treatment. Scale bars, 100 μm. **f** Expression levels of genes associated with lipid anabolism (*Acaca*, *Srebp2*, *Lpl*, and *Fasn*) and catabolism (*Acads*, *Cpt1a*, and *Acadvl*)

in gWAT from *SOST^ki* mice with or without LA5 treatment detected by qPCR. **g** Fasting blood glucose in *SOST^ki* mice with or without LA5 treatment. **h** Glucose metabolism in *SOST^ki* mice with or without LA5 treatment. Glucose tolerance test (GTT) (left), insulin tolerance test (ITT) (middle), and area under the curve (AUC) analysis for GTT and ITT (right). **i** Expression levels of genes associated with glucose metabolism in gWAT from *SOST^ki* mice with or without LA5 treatment detected by qPCR. Note: *n* = 6 biologically independent samples per group. All data were expressed as mean ± SD. \**P* < 0.05, \*\**P* < 0.01, \*\*\**P* < 0.001, \*\*\*\**P* < 0.0001 for inter-group comparison. Statistical significance was calculated using unpaired *t*-test. ns: no significance. All tests were two-sided. Mouse image was created in BioRender. Zhang, G. (2025) https://BioRender.com/17da8dl.

*Lrp4m/SOST^ko* mice with sclerostin overproduction were also observed compared to corresponding mice without sclerostin overproduction (Fig. 9i, Supplementary Fig 12d). The above genetic evidence indicated that *Lrp4* mutation-induced blockade of adipocytic sclerostin loop3-LRP4 interaction attenuated the impairment effect of sclerostin on lipid and glucose metabolism in vivo.

Third, to pharmacologically determine whether sclerostin loop3-LRP4 interaction participated in the impairment effect of sclerostin on whole-body lipid and glucose metabolism in vivo, the blocking peptide LA5 (10 mg/kg, once a day) that could block sclerostin loop3-LRP4 interaction was used to treat 6-month-old *SOST^ki* mice (Fig. 10a). Before formal experiments, the dosage of LA5 administration was determined in *SOST^ki* mice with a small sample size (*n* = 3) (Supplementary Fig 13). Similarly, the body weights remained unchanged in *SOST^ki* mice with and without LA5 treatment (Fig. 10b). In *SOST^ki* mice treated with LA5 (*SOST^ki* + LA5), the fat pad weights and serum FFA levels were lower than those in *SOST^ki* mice treated with Vehicle (*SOST^ki* + Veh) (Fig. 10c, d). The adipocytes in

both gWAT and iWAT of *SOST^ki* + LA5 group were smaller relative to those of *SOST^ki* + Veh group (Fig. 10e). The expression levels of genes associated with lipid anabolism were significantly lower, but the expression levels of genes associated with lipid catabolism were significantly higher in gWAT and iWAT of *SOST^ki* + LA5 group than those of *SOST^ki* + Veh group (Fig. 10f and Supplementary Fig. 14a). Despite the fact that the FBG level was comparable, the glucose tolerance and insulin sensitivity were improved in *SOST^ki* + LA5 group when compared to those in *SOST^ki* + Veh group (Fig. 10g, h). *SOST^ki* + LA5 group showed lower IRS1 phosphorylation but higher AKT phosphorylation in gWAT than the *SOST^ki* + Veh group (Supplementary Fig 14b). The expression levels of *Glut4* and *IRS1* in both gWAT and iWAT of *SOST^ki* + LA5 group were significantly lower when compared to those of *SOST^ki* + Veh group (Fig. 10i and Supplementary Fig 14c). The above pharmacologic evidence showed that the blocking peptide LA5-induced blockade of sclerostin loop3-LRP4 interaction attenuated the impairment effect of sclerostin on lipid and glucose metabolism in vivo.

## Discussion

For the first time in this study, it was found that sclerostin loop3 contributed to the impairment effect of sclerostin on lipid and glucose metabolism, in which the adipocytic sclerostin loop3-LRP4 interaction was required by sclerostin to impair whole-body lipid and glucose metabolism.

Previous clinical studies have shown that serum sclerostin levels in T2DM patients were significantly higher than those in healthy individuals[31,32]. Due to the multiple complications commonly associated with T2DM, T2DM patients often take medications other than anti-diabetes drugs for decades, which might induce remarkable fluctuations in serum sclerostin levels and, thereafter, misjudge the role of sclerostin in the pathogenesis of T2DM. By continuously tracking a large sample population from our longitudinal study, the significantly elevated serum sclerostin levels were found in patients with newly diagnosed T2DM. In combination with our cross-sectional data regarding POP patients with/without T2DM, it strongly indicated the potential of sclerostin as a biomarker and molecular target of T2DM.

In this study, both our genetic sclerostin loop3 deficiency and pharmacologic sclerostin loop3 inhibition data showed that sclerostin loop3 contributed to the impairment effect of sclerostin on lipid and glucose metabolism in vivo. Further, sclerostin loop3 was identified to interact with LRP4 in adipocytes. Both genetic studies by *Lrp4m*-based approaches and pharmacologic studies by LA5-based approaches consistently indicated that adipocytic sclerostin loop3-LRP4 interaction was required by sclerostin to promote lipid formation and glucose uptake in vitro, as well as impair lipid and glucose metabolism in vivo. This work bridged the understanding gap about how sclerostin impaired lipid and glucose metabolism.

The LRP4 in adipocytes was reported to impact the endocrine effects of sclerostin, thereby affecting whole-body lipid and glucose metabolism[29]. LRP4 has been found to bind to sclerostin in osteoblast, thereby inhibiting bone formation[30]. However, the question of how sclerostin interacted with LRP4 in adipocytes to participate in the endocrine effects of sclerostin was not addressed. For the first time, it was identified that sclerostin loop3 interacted with the fifth LA repeat (LA5) of LRP4 in adipocytes, thereby contributing to the impairment effect of sclerostin on lipid and glucose metabolism in our study.

Both POP patients and T2DM patients have higher cardiovascular risk than healthy individuals[33,34]. The marketed therapeutic antibody, mainly against sclerostin loop2, approved for treatment of POP was warned of cardiovascular risk, which limited its clinical application in POP patients with T2DM[9,35,36]. In our published work, sclerostin loop3 has become a precise bone anabolic target with no cardiovascular safety concern. Although the glucagon-like peptide-1 agonists (GLP-1 RAs) have become the "game changer" of obesity and T2DM treatment[37], it could not reverse the bone loss and even decrease bone mineral density in POP patients[38]. In our study, both semaglutide (GLP-1 RA) and sclerostin loop3-specific aptamer (Apc001OA) improved the lipid and glucose metabolism in vivo, whereas only Apc001OA significantly enhanced bone mass, improved bone microarchitecture, and bone mechanical properties in HFD-induced mice. Targeting sclerostin loop3 could be a potential strategy to normalize lipid and glucose metabolism, besides promoting bone formation without cardiovascular safety concern.

The clinical data of the marketed sclerostin antibody, mainly targeting loop2, demonstrated that the bone formation peaked at 6 months and decreased continuously thereafter[13], which could be explained by the compensatorily elevated DKK1 after treatment[39]. Both US-FDA and EMA limited the clinical use of the marketed sclerostin antibody within one year (FDA Press Announcements, European Medicines Agency Documents). POP and T2DM are chronic diseases, that need long-term treatment. In our study, it was found that the DKK1 level was not elevated in HFD-induced mice with Apc001OA treatment. Consistently, adipocytic LRP4 deficiency was determined to not

induce the compensatorily elevated DKK1 in Kim's study[29]. It implied that specific blockade of adipocytic sclerostin loop3-LRP4 interaction could not lead to the compensatorily elevated DKK1, which could be developed as a long-term therapeutic strategy.

The serum sclerostin levels were significantly higher in POP patients with T2DM and OVX mice fed a HFD compared to corresponding control group. In our recent study and collaborative research, the glucocorticoid signaling in bone tissue was found to be activated under metabolic stress conditions, such as high-fat diet induction[22,40]. This induced the expression and secretion of sclerostin from bone, which subsequently contributed to whole-body lipid and glucose metabolism disorders[40]. It suggests that bone is a vital organ for the regulation of lipid and glucose metabolism, in which bone-derived sclerostin could play a mediating role in the crosstalk between bone and other organs.

However, several limitations of this study should be acknowledged. First, other mouse models (e.g., db/db mouse) with lipid and glucose metabolism disorders should be further investigated. Second, we failed to obtain the adipocytic *Lrp4* mutation (*Ad.Lrp4m*) mice due to technical limitation. Alternatively, we constructed the global *Lrp4* mutation mice with *Lrp4m* in adipocytes being corrected to wild-type *Lrp4* (*Lrp4m/Ad-Lrp4*) to investigate the role of adipocytic *Lrp4* in lipid and glucose metabolism disorders. Third, the downstream signaling pathways of adipocytic sclerostin loop3-LRP4 interaction remain to be fully elucidated.

From the above, our study laid mechanistic foundation for the development of a long-term treatment strategy by blocking adipocytic sclerostin loop3-LRP4 interaction, to normalize lipid and glucose metabolism in POP patients coexisting with T2DM, besides targeting sclerostin loop3 to promote bone formation in cardiovascular safety.

## Methods

All animal experiments were conducted in compliance with approved protocols by the Chinese University of Hong Kong animal care committees. All research complies with relevant ethical regulations.

### Clinical data

The clinical data were obtained from consenting individuals at the Shanghai Tenth People's Hospital. All samples were collected ethically with clinical approval, and explicit written informed consent from each participating patient was obtained. (1) Study 1: This was a 5-year retrospective study conducted in the Shanghai region of China as part of the Diabetes-Osteoporosis Interaction Program. The study included 1007 participants without T2DM at baseline and had complete data by follow-up examinations. Blood glucose, serum free fatty acids (FFA) levels, and serum sclerostin levels were measured yearly since 2020. The HbA1c levels in newly diagnosed T2DM patients were ≥ 6.5%. The participants in this study were aged above 50 years. Subjects registered were permanent residents in community of Shanghai from 2020 to 2024. Among the 3500 participants followed up, 1969 participants were excluded because they already suffered from hyperglycemia or other diseases within the past 6 months; 312 participants were lost to follow-up, and 20 participants died; 192 participants were excluded because of incomplete data or lack of informed consent. The remaining 1007 participants (male: female = 458: 549) were included in the present study (HbA1c < 6.5%). Over the 5-year follow-up, 119 participants developed T2DM. (2) Study 2: 65 postmenopausal women were recruited in this project. Oral glucose tolerance test (OGTT) data were used to divide them into the normal glucose tolerance (NGT) group and the T2DM group. Among them, 19 postmenopausal women were diagnosed with normal bone mass (NBM), while 46 postmenopausal women were diagnosed with osteoporosis. All individuals were divided into four groups: NBM + NGT group, NBM + T2DM group, POP + NGT group, POP + T2DM group. Ethical approval was obtained

from the Ethics Committee of Shanghai Tenth People's Hospital (approval number: 24K110).

## Cell culture

3T3-L1 cells (ATCC CL-137) were cultured in 5% $CO_2$ and maintained in DMEM (ThermoFisher Scientific, 11965126) supplemented with 10% FBS and 1% penicillin-streptomycin Solution (preadipocyte medium). To induce adipocyte differentiation, differentiation medium was prepared by adding 0.5 mM methylisobutylxanthine, 1 μM dexamethasone, and 5 μg/mL insulin to preadipocyte medium. Two days after confluence (day 0), 3T3-L1 cells were induced in differentiation medium. On day 3, differentiation medium was replaced with maintenance medium (preadipocyte medium supplemented with 5 μg/mL insulin), which was changed every 2 days thereafter until analysis. For drug treatment, 10 nM sclerostin or blocking peptide LA5 (synthesized by GL Biochem Ltd., Shanghai) was used to treat cells when replacing medium at each time.

## Lentivirus transduction and puromycin selection

3T3-L1 cells were seeded at $2 \times 10^5$ cells per well in a final volume of 1 mL per well with a multiplicity of infection (MOI) of 60. Until 30–50% confluence, the lentivirus particles ($10^8$ Tu/mL) were added to the cells. 24 h later, the medium containing the remaining lentivirus was replaced with fresh medium. After two days, puromycin was used for the screening of cells successfully transduced with lentivirus packaging target plasmids. Before formal experiments, the optimal puromycin concentration was determined to be 2.4 μg/mL. Three days after screening, the remaining cells were ready for passaging.

## Plasmid construction

To construct recombinant plasmids encoding truncated LRP4 or LRP4 muteins, pcDNA 3.1-LRP4 was prepared as a template for amplification of the open reading frame by PCR (Supplementary Table 2). Supplementary Table 3 listed the sequences of LA5 domain in LRP4 muteins. The empty vector pcDNA 3.1 was linearized with restriction endonucleases EcoRV and XhoI. After the ligation of PCR product, DNA fragment of FLAG tag and linearized vector using a one-step cloning kit (Vazyme, C112-01), the ligation products (20 μL) were transformed into *E. coli* DH5α competent cells (100 μL), which were then incubated on agar plates for over 12 h. Colony PCR and restriction enzyme digestion were used to pick up positive clones for subsequent plasmid amplification and extraction[40].

## Luciferase reporter assay

The sclerostin knock-out 3T3-L1 cells were cultured in DMEM grown to 60–80% confluence, and transfected with *Wnt1*, plasmids encoding LRP4 muteins, TopFlash and Renilla control vectors using Lipofectamine 3000 Transfection Reagent (Vazyme, TL301-01). After 6 h of incubation, the medium containing transfection reagent was replaced with fresh medium. At this time, sclerostin (10 nM) or blocking peptide (10 nM) was added to treat cells. The cells were then incubated for 48 h and assayed for luciferase activity using Dual Luciferase Reporter Assay Kit (Promega, E1960) according to the manufacturer's instructions.

## Pull-down assay

Three days after transfection, cell lysates were collected in RIPA buffer containing protease inhibitor cocktail and centrifuged at $3000 \times g$ for 30 min at 4 °C to replace the RIPA solution with binding buffer (PBS, 20 mM imidazole, pH 8.0). After replacement with binding buffer three times, 25 μL Ni-NTA Magnetic Agarose Beads were incubated with 7.5 μg His-tagged sclerostin (His-sclerostin) and 7.5 μg His-tagged loop3 peptide (His-loop3 peptide) with gentle rotation for 2 h at 4 °C, respectively. Then, 500 μL of cell lysate supernatant was added to beads carrying His-sclerostin or His-loop3 peptide for incubation

overnight with gentle rotation at 4 °C. Following the incubation, 500 μL of wash buffer (PBS, 20 mM imidazole, 0.005% Tween 20 (v/v), pH 8.0) was used to wash the beads three times, and elution buffer (PBS, 250 mM imidazole, 0.005% Tween 20 (v/v), pH 8.0) was used to elute the bound proteins. To prepare the samples, the eluate was mixed with SDS-PAGE loading buffer and heated to 95 °C for 10 minutes. These samples were then subjected to SDS-PAGE and Western blot using antibodies that recognize target proteins[41].

## Western blot

The protein samples were electrophoresed on 8% bis-Tris gels after being denatured for 10 minutes at 95 °C. After the proteins were separated, they were transferred to 0.22 μm PVDF membranes (ThermoFisher Scientific, 88518) at a constant current of 350 mA for 4 hours. The membranes were then washed three times for 10 min each time in Tris-buffered saline with 0.1% Tween 20 (TBST). The membranes were blocked for an hour at room temperature in 5% dry skim milk. After that, membranes were probed overnight at 4 °C using primary antibodies, including anti-FLAG (Abcam, ab125243), anti-His (Abcam, ab245114), anti-p-IRS1 (Ser307; Cell Signaling Technology, 2381S), anti-IRS1 (Cell Signaling Technology, 2382S), anti-p-AKT (Ser473; Cell Signaling Technology, 9271S), anti-AKT (Cell Signaling Technology, 4691S), and anti-β-Actin (Cell Signaling Technology, 3700S). Following three washes with TBST, membranes were incubated for 1 h at room temperature with either HRP-linked anti-mouse IgG (Cell Signaling Technology, 7076, 1:10000) or HRP-conjugated anti-rabbit IgG (ThermoFisher Scientific, 31460, 1:10000). The ECL reagent (Bio-Rad, 1705061) was utilized to detect the protein bands. The Azure c300 Chemiluminescent Western Blot Imaging System was used to detect and record chemiluminescent signals.

## Biolayer interferometry (BLI)

LRP4 was biotinylated for 2 h at 4 °C using 10 mM EZ-Link Sulfo-NHS-LC-LC-Biotin (ThermoFisher Scientific, 21343) at a molar ratio of 1:20 (LRP4: biotin). The following procedures were performed to remove excess biotin using the Slide-A-Lyzer Dialysis Cassettes (ThermoFisher Scientific, 66382): (1) dialysis for 2 h at 4 °C; (2) dialysis buffer was changed and dialyzed for additional 2 h at 4 °C; (3) dialysis buffer was changed and dialyzed overnight at 4 °C. The volume of dialysis buffer was 200–500 times the sample volume. The Biomolecular Interaction Analyzer (ForteBio, Octet RED96e) was used to conduct BLI analysis. The His-tagged and biotin-labeled proteins were loaded onto anti-penta-HIS (HIS1K) and streptavidin (SA) Tray biosensors (Sartorius, 18-5019), respectively. In a 96-well plate, biosensors were pre-wetted with 200 μL PBS containing 0.02% Tween 20 (v/v) for 10–15 min. The biosensors were then immersed in sample solutions for 600 s at 25 °C with shaking at 300 rpm. Raw data were analyzed using ForteBio Data Analysis Software (v8.1). After subtracting non-specific binding between biosensors and samples, all $K_d$ values were calculated.

## Oil red O staining

Cells were first washed with PBS and then fixed with 4% paraformaldehyde (PFA) solution for 10 min. Cells were then washed twice with PBS, followed by covering with 60% isopropanol for 20 s. After removing 60% isopropanol, the cells were covered by enough Oil red O staining solution (Servicebio, G1015) for 10–30 min. Cells were then washed with 60% isopropanol for 30 s and covered with PBS for subsequent imaging of lipid droplets. Oil red O dye was eluted with 100% isopropanol for 5 min. Adipogenic differentiation was quantified by measuring absorbance at 520 nm using a spectrophotometer[42].

## Mice and genotyping

Animals in this study were obtained from the Laboratory Animal Services Center of the Chinese University of Hong Kong. The Animal Experimentation Ethics Committee of the Chinese University of Hong

Kong approved the experimental protocols (Ethical Permission NO: 23-212-MIS and 22-350-MIS). Mice were housed in a temperature-controlled environment with a 12-h light/12-h dark cycle, with ad libitum access to food and water. Female C57BL/6J mice (8 weeks old) underwent bilateral ovariectomy (OVX) or sham operation (Sham). Afterwards, mice were fed either a chow diet (Chow; 10% kcal from fat, 3.76 kcal/g; D12450J, Research Diets, Inc., USA) or a high-fat diet (HFD; 60% kcal from fat, 5.24 kcal/g; D12492, Research Diets, Inc., USA) for 12 weeks before sacrifice. Body weight was measured 12 weeks after surgery. For *SOST*[WT] (wild-type) and *SOST*[loop3m] C57BL/6J mice, 6-week-old male mice were fed a HFD for 16 weeks. For mice treated with saline/Apc001OA/semaglutide, 10-week-old male mice were fed a HFD for 12 weeks and then received treatment for 8 weeks[43]. The food consumption was recorded weekly. Energy intake was calculated according to the caloric value from food provided by Research Diets. After anesthesia with isoflurane, mice were sacrificed for sample collection.

The construction and genotyping of *SOST*[ki] C57BL/6J mice were as described in our previous work[19,20]. The male *SOST*[ki] mice were genotyped using the following primers: 5′-ATGCCCACCAAAGTCAT-CAGTGTAG-3′ and 5′-AGGCGGGCCATTTACCGTAAGTTA-3′ for the 5′ arm of *SOST* allele (1465-bp product), 5′-CCTCCTCTCCTGACTACTCCCAGTC-3′ and 5′-TCACAGAAACCATATGGCGCTCC-3′ for the 3′ arm of *SOST* allele (1229-bp product), and 5′-CAGCAAAACCTGGCTGTGGATC-3′ and 5′-ATGAGCCACCATGTGGGTGTC-3′ for wild-type (WT) allele (412-bp product). The male *Δloop3-SOST*[ki] C57BL/6J mice were genotyped using the following primers: 5′-ATGCCCACCAAAGTCATCAGTGTAG-3′ and 5′-AGGCGGGCCATTTACCGTAAGTTA-3′ for the 5′ arm of *SOST* allele (1465 bp), 5′-CTAGAGCCTCTGCTAACCATGTTC-3′ and 5′-TCACAGAAACCATATGGCGCTCC-3′ for the 3′ arm of *SOST* allele (825 bp), and 5′-CAGCAAAACCTGGCTGTGGATC-3′ and 5′-ATGAGCCACCATGTGGGTGTC-3′ for WT allele (412-bp product). The male *SOST*[loop3m] C57BL/6J mice were genotyped using the following primers: 5′-GACTACAAAGACCATGACGGTGATT-3′ and 5′-TACACTGAAAACGAATCGGATCGC-3′ for *SOST*[loop3m] allele (121 bp), and 5′-GGTGCCTCCTTCCTATAATCCATA-3′ and 5′-GAGGCCACCAGACGCACCTT-3′ for WT allele (398 bp). The *SOST*[ko] C57BL/6J mice were genotyped using the following primers: 5′-AGTGATATGGTGAGGCTGGATGC-3′ and 5′-GAACCTCAGTGATGGCTTAGTGG-3′ for *SOST*[ko] allele (555 bp), and 5′-ACACACAATGTCTCGCCACTGT-3′ and 5′-CAGCTAACTGAAGAGACAGGGATAG-3′ for WT allele (378 bp). The male *Lrp4m* C57BL/6J mice were genotyped using the following primers: 5′-TTGGGTGGACAGGCATCAATG-3′ and 5′-CCTCCCTGAATAACTTCGTATAATGTATGC-3′ for the 5′ arm of *Lrp4* allele (1567 bp), 5′-GATCCCCATCAAGCTGATAACATACG-3′ and 5′-GACACTCACGGCAGCTTTCCTC-3′ for the 3′ arm of *Lrp4* allele (1929 bp), and 5′-GGGAACGAAGCTACAACCTGGAC-3′ and 5′-CGACAGTCCTGCTCATCAGAGTC-3′ for WT allele (835 bp). The Adipoq-Cre C57BL/6J mice were genotyped using the following primers: 5′-GGATGTGCCATGTGAGTCTG-3′ and 5′-ACGGACAGAAGCATTTTCCA-3′ for *Cre* allele (200 bp) and 5′-CTATCAGGGATACTCCTCTTTGCC-3′ and 5′-GATACAGGAATGACAAGCTCATGGT-3′ for WT allele (507 bp) (Supplementary Fig 2). The obtained heterozygous *Lrp4m* mice were used for subsequent crossbreeding with Adipoq-Cre mice and *SOST*[ko] mice to obtain male *Lrp4m/Ad-Lrp4* mice and *Lrp4m/SOST*[ko] mice, respectively. As previously described by Kim et al., sclerostin overproduction was achieved by injecting adeno-associated viruses (AAV8-CMV) encoding mouse *SOST* (Vector Biolabs, AAV-272868) via the tail vein to 6-month-old male C57BL/6J mice for 6 weeks[10]. The blocking peptide (LA5, 10 mg/kg) was injected subcutaneously into 6-month-old male *SOST*[ki] mice once a week for 6 weeks to evaluate its effect on lipid and glucose metabolism.

## Glucose tolerance test (GTT) and Insulin tolerance test (ITT)
(1) For GTT, mice were fasted for 14 h and then administered glucose (2.0 g/kg body weight) via oral gavage. Blood glucose concentration was measured at 0, 15, 30, 60, 90, and 120 min over a 120-min period. (2) For ITT, mice were fasted for 4 h and then injected intraperitoneally with insulin (0.75 U/kg body weight). Blood glucose concentration was measured at 0, 15, 30, 60, 90, and 120 min over a 120-min period.

## Circulating parameters
(1) During GTT and ITT, blood was collected via tail vein. Blood glucose concentration was measured using a portable blood glucose apparatus and glucose testing strips. The fasting blood glucose (FBG) level was also measured. (2) Before euthanasia, blood samples were collected in heparin-coated capillary tubes. Serum free fatty acids (FFA) levels were measured using a colorimetric assay (Abcam, ab65341). Serum sclerostin levels were measured using ELISA kits (Cloud-Clone, SEC864Mu; R&D, DSST00) according to the manufacturers' protocols. All samples were analyzed in duplicate. The intra-assay coefficient of variation (CV) using the above kits was less than 10%, and all inter-assay CVs were below 12%[19,20].

## Hematoxylin and Eosin (HE) staining
Tissues were fixed in 4% PFA for more than 24 h, which were then dehydrated in sucrose (10%, 20%, 30% w/v) and embedded in paraffin. To detect the morphological changes in adipose tissues, 5-µm-thick dewaxed sections were stained with HE. They were then mounted for light microscopy analysis. The frequency distribution of adipocyte sizes was calculated by counting the number of adipocytes of different sizes using ImageJ[44].

## Quantitative real-time PCR (qPCR)
Total RNA was isolated using TRIzol (Invitrogen, 15596026) according to the manufacturer's instructions. After purification, total RNA was reverse transcribed into cDNA using the High-Capacity cDNA Reverse Transcription Kit (Applied Biosystems, 10249814). qPCR was performed using Quantitative SYBR Green PCR Kit (Vazyme, Q712). Data were calculated relative to the internal control (*β-Actin*) using the $2^{-\Delta\Delta CT}$ method. Data analysis was performed for acetyl-Coenzyme A carboxylase alpha (*Acaca*), sterol regulatory element binding factor 2 (*Srebp2*), lipoprotein lipase (*Lpl*), fatty acid synthase (*Fasn*), short-chain acyl-Coenzyme A dehydrogenase (*Acads*), carnitine palmitoyl-transferase 1a (*Cpt1a*), very long-chain acyl-Coenzyme A dehydrogenase (*Acadvl*), and *β-Actin*. All primers were synthesized by IGE Biotechnology (Guangzhou, China), and primers used in this study are listed in Supplementary Table 4.

## Micro-CT analysis
In mouse skeletal tissues, bone mass and microarchitecture integrity of trabecular bone at the right distal femur and proximal tibia, as well as cortical bone at the right femoral mid-shaft, were analyzed using micro-computed tomography (micro-CT, version 6.5). The region of the distal femur extending distally along the femoral diaphysis from the growth plate, the region of the femoral mid-shaft, and the region of the proximal tibia extending proximally along the tibial diaphysis from the growth plate were all scanned using 319 slices with voxels of 10 µm. Regions of interest (ROIs) were established using the Scanco evaluation software to calculate trabecular and cortical parameters. One hundred consecutive slices starting at 0.1 mm from the most proximal portion of the growth plate, in which both condyles were no longer visible, were selected for analysis of the distal femur and proximal tibia. By physically sculpting the trabeculae, the cortical bone was excluded for analysis. To ensure ROIs were within the endosteal envelope, freehand ROIs of the trabeculae were drawn on 100 consecutive slices. Using the automated thresholding algorithm, 100 consecutive slices of the femoral mid-shaft were analyzed at both the distal 50% of the femur length and its precise center[19,20].

## Three-point bending

Each specimen was subjected to a three-point bending test to failure using a universal testing machine (H25KS; Hounsfield Test Equipment Ltd., UK). Femurs were loaded anterior-posterior for the three-point bending test, with a span of 17 mm. At the mid-shaft of femur, a load was applied at a constant displacement rate of 1 mm/min. Throughout each experiment, the load and displacement of the loading device were recorded until the specimen fractured. Stiffness (N/mm) was determined by calculating the slope of the first linear uploading section of the load-displacement curves. From the load data, maximum failure load (failure force, N) values were obtained. Fracture energy was determined by calculating the area under the load-displacement curve until the specimen fractured[45].

## Statistical analysis

All numerical data were expressed as mean ± standard deviation (SD). Paired *t*-test, unpaired *t*-test, and two-way ANOVA with Sidak's multiple comparisons tests were performed to evaluate intergroup differences using GraphPad Prism 9. $P < 0.05$ was considered statistically significant.

## Reporting summary

Further information on research design is available in the Nature Portfolio Reporting Summary linked to this article.

## Data availability

Source data are provided with this paper. The raw data for the main text and supplementary information have been made available as downloadable Excel files. Source data are provided with this paper.

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

## Acknowledgements

This study was supported by the direct grant of The Chinese University of Hong Kong (Project No. 4054660), National Key R&D Program of China (No. 2018YFA0800804), Hong Kong General Research Fund from the Research Grants Council of the Hong Kong Special Administrative Region, China (Project Nos. 12100725, 12100921, 12102120, 12102223, 12102524, 14103121, 14103420 and 14109721), Theme-Based Research Scheme from the Research Grants Council of the Hong Kong Special Administrative Region, China (Project No. T12-201/20-R), Young Scientists Fund of the National Natural Science Foundation of China (Grant No. 82300988), Shenzhen-Hong Kong-Macau Science and Technology Plan Project (Category C) (Grant No. SGDX20230821095359002), Basic and Applied Basic Research Fund from Department of Science and Technology of Guangdong Province (Project No. 2019B1515120089), Inter-institutional Collaborative Research Scheme from Hong Kong Baptist University (Project No. RC-ICRS/19-20/01), University-Industry Collaboration Programme from Innovation and Technology Commissions of the Hong Kong Special Administrative Region, China (Project No. UIM/298), University-Industry Collaboration Programme from Innovation and Technology Commissions of the Hong Kong Special Administrative Region, China (Project No. UIM/328), Key Project of Research and Development Plan of Hunan Province (Project No. 2022WK2010), and Youth's Project of Guangdong Basic and Applied Basic Research Fund (GDSTC No. 2022A1515110044).

## Author contributions

B.-T.Z., G.Z., L.W., D.L., and H.S. supervised the whole project. H.J., X.T., S.Y., Y.Z., and Y.M. performed the major research, analyzed the data, and participated in manuscript writing and revision with equal contributions. N.L., S.W., N.Z., X.Y., and S.D. conducted the genotyping and bone metabolism analysis. C.Z., H.L., Z.L., X.W., H.Z., Z.C., and M.S. provided technical support for in vitro and in vivo studies. H.L., M.R., C.L., Y.M., Y.Y., J.L., Z.Z., and A.L. provided their professional expertise in aptamer-based drug discovery and manuscript revision.

## Competing interests

The authors declare no competing interests.
