## [Transparent Peer Review file · Nature Communications]

Adipocytic sclerostin loop3-LRP4 interaction required by sclerostin to impair whole-body lipid and glucose metabolism

Corresponding Author: Professor Baoting Zhang

Version 0:

Reviewer comments:

Reviewer #1

(Remarks to the Author)

The authors present a translational research study using clinical data from a cohort to establish the effect of sclerostin on lipid and glucose metabolism. They also look for possible therapeutic solutions.

The introduction is not typical. Initially, the basis for the subsequent establishment of the hypothesis is indicated. However, in the final part of the introduction, the possible results are discussed, which is not the usual procedure. On the other hand, on page 3, lines 78-80, data are given in relation to GLP-RA, which is not the objective of the study. The reason for including this data should be indicated.

The results are clearly expressed and follow an interesting pathophysiological order.

The discussion is short but adapted to the results obtained. The strengths and weaknesses of the study should be indicated. The methodology is exhaustively described but the technique used for the determination of sclerostin as well as its inter- and intra-assay coefficients of variation should be indicated.

Reviewer #2

(Remarks to the Author)

Regulation of bone and both lipid and glucose metabolism is of crucial interest for the management of chronic disease such as osteoporosis and diabetes. This study highlights the role of sclerostin loop3 in the impairment of lipid and glucose metabolism through in vivo tests by both genetic loop3 deficiency and pharmacologic loop3 inhibition. The authors found that sclerostin loop3 interacts with LRP4, in adipocytes. Both genetic studies by Lrp4m-based approaches and pharmacologic studies by LA5-based approaches consistently suggested adipocytic sclerostin loop3-LRP4 interaction was required by sclerostin to promote lipid formation and glucose uptake in vitro, as well as impair lipid and glucose metabolism in vivo.

The research study is well-conducted and results are consistent with the hypotheses.

The findings may have relevant clinical impact.

Reviewer #3

(Remarks to the Author)

This manuscript by Jiang et al. provides substantial evidence demonstrating the role of sclerostin loop3-LRP4 interaction in impairing systemic metabolism. This work timely expands current knowledges about sclerostin in regulation of metabolic homeostasis. The methodology is sound, and the conclusion is eligible.

I have some minor comments.

-In this study, insulin sensitivity was only evaluated by ITT assay. Moreover, the insulin signaling pathway, including the p-IRS and p-AKT levels of adipose tissue, should also be analyzed.

-Since inhibition of sclerostin loop3 exhibited systemically metabolic improvement, how about the alteration of insulin sensitivity in skeletal muscle?

-The abstract needs to be refined.

-In page 4, results related to figure 1 and supplementary figure 1 can be integrated into one section.

-It is interesting to determine the effect of sclerostin loop3 inhibition in diabetic animal models, such as db/db mice.

-The scale bars in many histological figures (fig2d,3e,4f,5h,8e,9e,10e) are hard to see.

Reviewer #4

(Remarks to the Author)

In the present manuscript, Jiang et al. investigate the role of sclerostin in whole-body lipid and glucose metabolism, with a particular focus on its Loop3 region. Through a series of experiments involving genetically modified mouse models, pharmacological inhibitors, and both in vitro and in vivo studies, the authors provide an extensive analysis of how sclerostin impacts bone metabolism and energy homeostasis. Key findings indicate that elevated levels of sclerostin are associated with impaired lipid and glucose metabolism in postmenopausal osteoporosis (POP) patients with type 2 diabetes mellitus (T2DM) and in ovariectomized mice fed a high-fat diet. The study demonstrates that sclerostin interacts with LRP4 through its Loop3 region to influence metabolic processes. They compared wild-type mice, full-length sclerostin knock-in (SOSTki) mice, and Loop3-deficient sclerostin knock-in (Δ loop3-SOSTki) mice across multiple parameters such as body weight, fat pad weight, serum free fatty acid concentrations, and glucose uptake assays. Additionally, they synthesized and tested a blocking peptide LA5, which specifically targets the sclerostin Loop3-LRP4 interaction, further confirming the importance of this domain. Overall, the manuscript is well written, and most of the experiments are of high quality and support the author's claims.

The manuscript identifies that serum sclerostin levels are significantly higher in response to a high-fat diet, especially in the context of postmenopausal osteoporosis (POP) patients with type 2 diabetes mellitus (T2DM) and ovariectomized mice fed a high-fat diet. However, the underlying mechanism remains unclear. Is there a cyclic mechanism wherein glycolipid metabolism disorders first affect bone metabolism, and this change subsequently exacerbates the disordered state of glycolipid metabolism, with sclerostin (SOST) playing a mediating role?

In Supplementary Figure 1, why is there no significant difference in body weight between the ovariectomized (OVX) group and the control (chow) group? Theoretically, ovariectomy typically leads to obesity. At what time point after the surgery were the body weights measured?

Does SOST-loop3 regulate glucose and lipid metabolism solely through Lrp4, or are other receptors, such as LRP5/6, also involved in this process?

The clinical study section lacks relevant ethical statements. Were these studies approved and supported by the hospital's ethics committee?

The retrospective study section does not provide detailed statements on the inclusion and exclusion criteria for the study subjects. Please provide these details to better understand the study design.

Images of HE staining and Oil Red O staining should include scale bars to allow for accurate interpretation of the structures and scales within the images.

Version 1:

Reviewer comments:

Reviewer #1

(Remarks to the Author)

The questions have been answered by the authors

Reviewer #3

(Remarks to the Author)

The authors have addressed all my concerns. I have no more comments.

Reviewer #4

(Remarks to the Author)

The authors have addressed all the concerns I brought up in my original review and I have no further questions

Point-by-point response to the comments from reviewer #1

Number	Comments	Response	Position of Revision
	Overall comments: The authors present a translational research study using clinical data from a cohort to establish the effect of sclerostin on lipid and glucose metabolism. They also look for possible therapeutic solutions. The introduction is not typical. Initially, the basis for the subsequent establishment of the hypothesis is indicated. However, in the final part of the introduction, the possible results are discussed, which is not the usual procedure. On the other hand, on page 3, lines 78-80, data are given in relation to GLP-RA, which is not the objective of the study. The reason for including this data should be indicated. The results are clearly expressed and follow an interesting pathophysiological order. The discussion is short but adapted to the results obtained. The strengths and weaknesses of the study should be indicated. The methodology is exhaustively described but the technique used for the determination of sclerostin as well as its inter- and intra-assay coefficients of variation should be indicated.	Thank you very much for your insightful comments on our study. Below is our point-by-point response to each comment.	
1	The introduction is not typical. Initially, the basis for the subsequent establishment of the hypothesis is indicated. However, in the final part of the introduction, the possible results are discussed, which is not the usual procedure.	Thanks for your comments. We removed the experimental results and added a brief description of research design in the final part of the introduction as follows: In this study, we investigated the role of sclerostin loop3 in the impairment effect of sclerostin on whole-body lipid and glucose	Lines: 88-96

		metabolism using both genetic and pharmacologic approaches. Then, we investigated whether genetic loop3 mutation or pharmacologic loop3 inhibition could attenuate the HFD-induced whole-body lipid and glucose metabolism disorders. To mechanistically reveal how sclerostin loop3 participated in the impairment effect of sclerostin on whole-body lipid and glucose metabolism, we performed protein-protein interaction assays to identify a receptor of sclerostin loop3 in adipocyte. Further, we investigated whether the adipocytic sclerostin loop3-receptor (the low-density lipoprotein receptor-related protein 4, LRP4) interaction was required by sclerostin to impair whole-body lipid and glucose metabolism in vitro and in vivo.	
2	On the other hand, on page 3, lines 78-80, data are given in relation to GLP-RA, which is not the objective of the study. The reason for including this data should be indicated.	We agreed with your point that GLP1-RAs are not our objective of the study, so we removed the sentences on page 3, lines 78-80. Since GLP-1-RAs are current first-line drugs for type 2 diabetes mellitus, here we used GLP-1RA as a positive control in experimental design. We provided the following description: Semaglutide, a first-line drug for T2DM ^{26, 27, 28}, was used as a positive control.	Line 70 Line 250-251
3	The discussion is short but adapted to the results obtained. The strengths and weaknesses of the study should be indicated.	Thanks for your valuable suggestion. Agreed with your point and added descriptions of weaknesses as follows: However, several limitations of this study should be acknowledged. First, other mouse models (e.g. db/db mouse) with lipid and glucose metabolism disorders should be further investigated. Second, we failed to obtain the adipocytic Lrp4 mutation (Ad.Lrp4m) mice due to technical limitation. Alternatively, we constructed the global Lrp4 mutation mice with Lrp4m in adipocytes being corrected to wild-type Lrp4 (Lrp4m/Ad-Lrp4) to investigate	Lines: 524-530

		the role of adipocytic Lrp4 in lipid and glucose metabolism disorders. Third, the downstream signaling pathways of adipocytic sclerostin loop3-LRP4 interaction remain to be fully elucidated.	
4	The methodology is exhaustively described but the technique used for the determination of sclerostin as well as its inter- and intra-assay coefficients of variation should be indicated.	Thanks for your suggestion. In the methodology section, we added a description as follows: Serum sclerostin levels were measured using ELISA kits (SEC864Mu, Cloud-Clone; DSST00, R&D) according to the manufacturers' protocols. All samples were analyzed in duplicate. The intra-assay coefficient of variation (CV) using the above kits was less than 10%, and all inter-assay CVs were below 12%^{20,21}.	Line 685-688

Point-by-point response to the comments from reviewer #2

Number	Comments	Response	Position of Revision
	Overall comments: Regulation of bone and both lipid and glucose metabolism is of crucial interest for the management of chronic disease such as osteoporosis and diabetes. This study highlights the role of sclerostin loop3 in the impairment of lipid and glucose metabolism through in vivo tests by both genetic loop3 deficiency and pharmacologic loop3 inhibition. The authors found that sclerostin loop3 interacts with LRP4, in adipocytes. Both genetic studies by Lrp4m-based approaches and pharmacologic studies by LA5-based approaches consistently suggested adipocytic sclerostin loop3-LRP4 interaction was required by sclerostin to promote lipid formation and glucose uptake in vitro, as well as impair lipid and glucose metabolism in vivo. The research study is well-conducted and results are consistent with the hypotheses. The findings may have relevant clinical impact.	Thank you very much for your recognition and insightful comments on our study.	

Point-by-point response to the comments from reviewer #3

Number	Comments	Response	Position of Revision
	Overall comments: This manuscript by Jiang et al. provides substantial evidence demonstrating the role of sclerostin loop3-LRP4 interaction in impairing systemic metabolism. This work timely expands current knowledges about sclerostin in regulation of metabolic homeostasis. The methodology is sound, and the conclusion is eligible. I have some minor comments.  -In this study, insulin sensitivity was only evaluated by ITT assay. Moreover, the insulin signaling pathway, including the p-IRS and p-AKT levels of adipose tissue, should also be analyzed. -Since inhibition of sclerostin loop3 exhibited systemically metabolic improvement, how about the alteration of insulin sensitivity in skeletal muscle? -The abstract needs to be refined. -In page 4, results related to figure 1 and supplementary figure 1 can be integrated into one section. -It is interesting to determine the effect of sclerostin loop3 inhibition in diabetic animal models, such as db/db mice. -The scale bars in many histological figures (fig2d,3e,4f,5h,8e,9e,10e) are hard to see. 	Thank you very much for your insightful comments on our study. Below is our point-by-point response to each comment.	

1.	In this study, insulin sensitivity was only evaluated by ITT assay. Moreover, the insulin signaling pathway, including the p-IRS and p-AKT levels of adipose tissue, should also be analyzed.	According to your suggestion, we detected the protein expression levels of p-IRS1 and p-AKT in adipose tissue to consolidate our findings. The data are consistent with the ITT assays. In Results section, we added descriptions of the corresponding data as follow: Consistently, we observed a higher phosphorylation level of IRS1 and a lower phosphorylation level of AKT in gWAT of SOST^{ki} mice than WT mice, while Δloop3-SOST^{ki} mice showed a lower phosphorylation level of IRS1 and a higher phosphorylation level of AKT than SOST^{ki} mice (Supplementary Fig 3c). SOST^{ki} + Apc0010A group showed lower IRS1 phosphorylation but higher AKT phosphorylation in gWAT than the SOST^{ki} + Veh group (Supplementary Fig 4c). SOST^{loop3m} + HFD group showed lower IRS1 phosphorylation but higher AKT phosphorylation in gWAT than the SOST^{WT} + HFD group (Supplementary Fig 5b). Both HFD + Apc0010A group and HFD + Semaglutide group showed lower IRS1 phosphorylation but higher AKT phosphorylation in gWAT than the HFD + Vehicle group (Supplementary Fig 7d). We observed a lower phosphorylation level of IRS1 and a higher phosphorylation level of AKT in gWAT of Lrp4m mice compared to WT mice, while Lrp4m/Ad-Lrp4 mice showed a higher phosphorylation level of IRS1 and a lower phosphorylation level of AKT than Lrp4m mice (Supplementary Fig 11b). Sclerostin overproduction increased the IRS1 phosphorylation but decreased the AKT phosphorylation in gWAT of SOST^{ko} mice, whereas no such effects were observed in Lrp4m/SOST^{ko} mice	Lines: 152-155 Lines: 186-188 Lines: 217-219 Lines: 270-272 Lines: 410-413 Lines: 437-439
----	--	--	---

		(Supplementary Fig 12c). SOST^{ki} + LA5 group showed lower IRS1 phosphorylation but higher AKT phosphorylation in gWAT than the SOST^{ki} + Veh group (Supplementary Fig 14b).	Lines: 457-459
2.	Since inhibition of sclerostin loop3 exhibited systemically metabolic improvement, how about the alteration of insulin sensitivity in skeletal muscle?	Thanks for your comments. According to your suggestion, we detected the protein expression levels of p-IRS1 and p-AKT in muscle of HFD-induced mice with/without Apc0010A (sclerostin loop3-specific aptamer) treatment. In Results section, we added a description of the data: Interestingly, HFD + Apc0010A group also showed lower IRS1 phosphorylation but higher AKT phosphorylation in skeletal muscle than the HFD + Vehicle group (Supplementary Fig 7e). It indicates that targeting sclerostin loop3 could enhance insulin sensitivity in skeletal muscle.	Lines: 272-274
3.	The abstract needs to be refined.	Thank you for pointing this out. We have addressed this issue by refining the mechanistic part in the abstract as follows (highlighted in yellow): New version Sclerostin, a typical negative regulator of bone formation, impairs whole-body lipid and glucose metabolism. Our findings showed that elevated serum sclerostin levels were found in both postmenopausal osteoporosis (POP) patients with T2DM and ovariectomized mice fed a high-fat diet (HFD). Surprisingly, the serum sclerostin levels were significantly elevated in newly diagnosed type 2 diabetes mellitus (T2DM) patients in our large-sample follow-up analysis. Sclerostin has three loops, including loop1, loop2 and loop3. The marketed therapeutic sclerostin antibody mainly targeting loop2 promoted bone formation and improved whole-body lipid and glucose metabolism. However, the FDA and EMA have warned its cardiovascular risk in	

		POP patients. We previously demonstrated that sclerostin loop3 contributes to the inhibitory effect of sclerostin on bone formation, whereas the cardiovascular protective action of sclerostin is independent of loop3. It is of great interest to investigate whether and how sclerostin loop3 participates in the impairment effect of sclerostin on whole-body lipid and glucose metabolism. This could facilitate developing next-generation sclerostin inhibitor without cardiovascular safety concern for POP patients with T2DM. In this study, both genetic loop3 deficiency and pharmacologic loop3 inhibition by Apc001OA (a sclerostin loop3-specific aptamer) attenuated the impairment effect of sclerostin on whole-body lipid and glucose metabolism in vivo. Both genetic loop3 mutation and pharmacologic loop3 inhibition by Apc001OA consistently attenuated HFD-induced whole-body lipid and glucose metabolism disorders. Mechanistically, both Lrp4 mutation and pharmacologic peptide tool (LRP4 blocking peptide LA5)-induced specific blockade of sclerostin loop3-LRP4 interaction was found to attenuate the impairment effect of sclerostin on lipid and glucose metabolism in vitro and in vivo. This study provided an innovative strategy, blocking adipocytic sclerostin loop3-LRP4 interaction, to normalize lipid and glucose metabolism in POP patients coexisting with T2DM, besides targeting sclerostin loop3 to promote bone formation in cardiovascular safety.	Lines: 49-52
4.	In page 4, results related to figure 1 and supplementary figure 1 can be integrated into one section.	Thanks for your kind suggestion. In the manuscript, the title and results related to Figure 1 and Supplementary Figure 1 have been integrated into one section: New title: Serum sclerostin levels were significantly higher in newly diagnosed T2DM (ND-T2DM) patients, POP patients with T2DM and corresponding animal models.	Lines: 99-100

5.	It is interesting to determine the effect of sclerostin loop3 inhibition in diabetic animal models, such as db/db mice.	Thanks for your comments. This is one weakness of our study. db/db mouse is indeed an extensively used non-diet-induced diabetic animal model in the field of lipid and glucose metabolism. In the future, we will further determine the effect of sclerostin loop3 inhibition in other models with lipid and glucose metabolism disorders. In the Discussion section, we added descriptions of several limitations of this study as follows: However, several limitations of this study should be acknowledged. First, other mouse models (e.g., db/db mouse) with lipid and glucose metabolism disorders should be further investigated.	Lines: 524-525
6.	The scale bars in many histological figures (fig2d,3e,4f,5h,8e,9e,10e) are hard to see.	Thanks for your suggestions. We updated these figures to make it clear for readers to see.	

Point-by-point response to the comments from reviewer #4

Number	Comments	Response	Position of Revision
	Overall comments: In the present manuscript, Jiang et al. investigate the role of sclerostin in whole-body lipid and glucose metabolism, with a particular focus on its Loop3 region. Through a series of experiments involving genetically modified mouse models, pharmacological inhibitors, and both in vitro and in vivo studies, the authors provide an extensive analysis of how sclerostin impacts bone metabolism and energy homeostasis. Key findings indicate that elevated levels of sclerostin are associated with impaired lipid and glucose metabolism in postmenopausal osteoporosis (POP) patients with type 2 diabetes mellitus (T2DM) and in ovariectomized mice fed a high-fat diet. The study demonstrates that sclerostin interacts with LRP4 through its Loop3 region to influence metabolic processes. They compared wild-type mice, full-length sclerostin knock-in (SOST^{ki}) mice, and Loop3-deficient sclerostin knock-in (Δloop3-SOST^{ki}) mice across multiple parameters such as body weight, fat pad weight, serum free fatty acid concentrations, and glucose uptake assays. Additionally, they synthesized and tested a blocking peptide LA5, which specifically targets the sclerostin Loop3-LRP4 interaction, further confirming the importance of this domain. Overall, the manuscript is well written, and most of the experiments are of high quality and support the author's claims. The manuscript identifies that serum sclerostin levels are significantly higher in response to a high-fat diet, especially in the context of postmenopausal osteoporosis (POP) patients with type 2 diabetes mellitus (T2DM) and ovariectomized mice fed a high-fat diet. However, the	Thank you very much for your insightful comments on our study. Below is our point-by-point response to each comment.	

	underlying mechanism remains unclear. Is there a cyclic mechanism wherein glycolipid metabolism disorders first affect bone metabolism, and this change subsequently exacerbates the disordered state of glycolipid metabolism, with sclerostin (SOST) playing a mediating role? In Supplementary Figure 1, why is there no significant difference in body weight between the ovariectomized (OVX) group and the control (chow) group? Theoretically, ovariectomy typically leads to obesity. At what time point after the surgery were the body weights measured? Does SOST-loop3 regulate glucose and lipid metabolism solely through Lrp4, or are other receptors, such as LRP5/6, also involved in this process? The clinical study section lacks relevant ethical statements. Were these studies approved and supported by the hospital's ethics committee? The retrospective study section does not provide detailed statements on the inclusion and exclusion criteria for the study subjects. Please provide these details to better understand the study design. Images of HE staining and Oil Red O staining should include scale bars to allow for accurate interpretation of the structures and scales within the images.		
1.	The manuscript identifies that serum sclerostin levels are significantly higher in response to a high-fat diet, especially in the context of postmenopausal osteoporosis (POP) patients with type 2 diabetes mellitus (T2DM) and ovariectomized mice fed a high-fat diet. However, the underlying mechanism	Thanks for your comments. Your outstanding explanation provided us with deep molecular insight into the underlying mechanism. According to our recently published study (Zhong C, et al. Targeting osteoblastic 11β-HSD1 to combat high-fat diet-induced	

	remains unclear. Is there a cyclic mechanism wherein glycolipid metabolism disorders first affect bone metabolism, and this change subsequently exacerbates the disordered state of glycolipid metabolism, with sclerostin (SOST) playing a mediating role?	bone loss and obesity. Nat. Commun. 2024) and collaborative research (Kim S, et al. Osteoblastic glucocorticoid signaling exacerbates high-fat-diet-induced bone loss and obesity. Bone Res. 2021), the glucocorticoid signaling in bone tissue was activated under metabolic stress conditions, such as high-fat diet induction. This induced the expression and secretion of sclerostin from bone, which subsequently contributed to whole-body lipid and glucose metabolism disorders. In the Discussion section, we added an explanation as follows: The serum sclerostin levels were significantly higher in POP patients with T2DM and OVX mice fed a HFD compared to corresponding control group. In our recent study and collaborative research, the glucocorticoid signaling in bone tissue was found to be activated under metabolic stress conditions, such as high-fat diet induction^{23,41}. This induced the expression and secretion of sclerostin from bone, which subsequently contributed to whole-body lipid and glucose metabolism disorders⁴¹. It suggests that bone is a vital organ for the regulation of lipid and glucose metabolism, in which bone-derived sclerostin could play a mediating role in the crosstalk between bone and other organs.	Lines: 516-522
2.	In Supplementary Figure 1, why is there no significant difference in body weight between the ovariectomized (OVX) group and the control (chow) group? Theoretically, ovariectomy typically leads to obesity. At what time point after the surgery were the body weights measured?	Thanks for your questions. Body weight was measured 12 weeks after surgery. One of the possible reasons could be explained by the sample size (n = 6) in this study. Although no significant difference in body weight was observed, there was a strong trend (P = 0.0792) toward higher body weight. In the Methodology section, we added a description as follows:	

		Body weight was measured 12 weeks after surgery.	Lines: 638-639
3.	Does SOST-loop3 regulate glucose and lipid metabolism solely through Lrp4, or are other receptors, such as LRP5/6, also involved in this process?	We sincerely appreciate your insightful comment. In our study, sclerostin loop3 was found not to bind with LRP6 directly. However, whether sclerostin loop3 could interact with other proteins to regulate lipid and glucose metabolism remains unclear. It is interesting to explore other binding receptors for sclerostin loop3 in future study. In the Results section, we added a description as follows: The sclerostin loop3 did not bind with LRP6 (Supplementary Figure 9b).	Lines: 317
4.	The clinical study section lacks relevant ethical statements. Were these studies approved and supported by the hospital's ethics committee?	Thanks for your comments. The ethics committee has approved our clinical studies, and the ethics application number has been added in the Methodology section as follows: Ethical approval was obtained from the Ethics Committee of Shanghai Tenth People's Hospital (approval number: 24K110).	Line 555-556
5.	The retrospective study section does not provide detailed statements on the inclusion and exclusion criteria for the study subjects. Please provide these details to better understand the study design.	Thanks for your concerns. The detailed statements on the inclusion and exclusion criteria for the retrospective study have been updated in the manuscript for better understanding. Inclusion criteria:  1. The participants in this study were aged above 50 years. 2. The baseline HbA1c levels in the participants were < 6.5%, while those of newly diagnosed T2DM patients were ≥ 6.5%. 	

		Exclusion criteria: 1. Participants suffering from hyperglycemia or other diseases within the past 6 months. 2. Participants who were lost to follow-up or died. 3. Participants with incomplete data or who lacked informed consent.	
6.	Images of HE staining and Oil Red O staining should include scale bars to allow for accurate interpretation of the structures and scales within the images.	Thanks for your comments. All the images have been updated in the manuscript.